# Equivalent current dipole sources of neurofeedback training-induced alpha activity through temporal/spectral analytic techniques

**Jen-Jui Hsueh[1], Yan-Zhou Chen[2], Jia-Jin Chen[3], Fu-Zen Shaw[1,2]\***

**1** Mind Research and Imaging Center, National Cheng Kung University, Tainan, Taiwan, **2** Department of Psychology, National Cheng Kung University, Tainan, Taiwan, **3** Department of Biomedical Engineering, National Cheng Kung University, Tainan, Taiwan

\* fzshaw@gmail.com

## Abstract

Much of the work in alpha NFT has focused on evaluating changes in alpha amplitude. However, the generation mechanism of training-induced alpha activity has not yet been clarified. The present study aimed to identify sources of training-induced alpha activity through four temporal/spectral analytic techniques, i.e., the max peak average (MPA), positive average (PA), negative average (NA) and event-related spectral perturbation average (ERSPA) methods. Thirty-five healthy participants were recruited into an alpha group receiving feedback of 8–12-Hz amplitudes, and twenty-eight healthy participants were recruited into a control group receiving feedback of random 4-Hz amplitudes from the range of 7 to 20 Hz. Twelve sessions were performed within 4 weeks (3 sessions per week). The control group had no change in the amplitude spectrum. In contrast, twenty-nine participants in the alpha group showed significant alpha amplitude increases exclusively and were identified as "responders". A whole-head EEG was recorded for the "responders" after NFT. The epochs of training-induced alpha activity from whole-head EEG were averaged by four different methods for equivalent current dipole source analysis. High agreement and Cohen's kappa coefficients on dipole source localization between each method were observed, showing that the dipole clusters of training-induced alpha activity were consistently located in the precuneus, posterior cingulate cortex (PCC) and middle temporal gyrus. The residual variance (goodness of fit) for dipole estimation of the MPA was significantly smaller than that of the others. Our findings indicate that the precuneus, PCC and middle temporal gyrus play important roles in enhancing training-induced alpha activity. The four averaging methods (especially the MPA method) were suitable for investigating sources of brainwaves. Additionally, three dipoles can be used for dipole source analysis of training-induced alpha activity in future research, especially the training sites are around the central regions.

**Editor:** Daqing Guo, Key Laboratory for NeuroInformation of Ministry of Education, School of Life Science and Technology, University of Electronic Science and Technology of China, CHINA

**Data Availability Statement:** All relevant data are within the manuscript.

**Funding:** This study was supported by the Ministry of Science and Technology, Taiwan (MOST 109-2627-H-006-005 | Recipient: FZS) (MOST 109-2634-F-006-013 | Recipient: FZS) (MOST 108-2410-H-006-112-MY3 | Recipient: JJH).

**Competing interests:** The authors have declared that no competing interests exist.

## Introduction

Neurofeedback training (NFT) is a noninvasive operant conditioning technique for training participants to self-regulate their own brain activity. Electroencephalogram (EEG) has been used most often to extract particular brainwave(s) as feedback from electrodes placed on the scalp. The purpose of NFT is to enable an individual to learn to self-regulate brain activity (a specific frequency band), which is supposed to be associated with specific behavior or state. In the last few decades, much of the work in alpha NFT has been focused on evaluating changes of alpha amplitude [1, 2]. Studies have reported that participants learned successfully to enhance their alpha amplitudes after several training sessions. The generation mechanism of training-induced alpha activity would be an interesting way to reveal possible brain processing when enhancing alpha activity.

Of the available tools for exploring the generation mechanism of training-induced alpha activity, EEGs have the advantages of noninvasiveness, portability and fine temporal resolution. Rhythmic alpha activity of the brain can be revealed by topographic EEG analysis. Previous studies have shown that training-induced alpha activity appeared diversely over the parieto-occipital [2], fronto-parietal [3], or frontal regions [4] through topographic EEG analysis. However, some studies did not report relevant information [5–7]. These discrepancies and uncertainties may make it difficult to understand sources of training-induced alpha activity and to provide possible explanation for improved cognition after alpha NFT. To our knowledge, sources of training-induced alpha activity after NFT have not been investigated recently. A systematic evaluation of the sources of training-induced alpha activity after NFT through dipole source analysis is desirable.

Dipole source localization provides a good opportunity to determine the generator of training-induced alpha activity. The greatest concern of dipole source localization is the signal-to-noise ratio (SNR) problem. A good SNR can decrease the estimation error of dipole source reconstruction [8]. Therefore, an available way to increase the SNR is through averaging; for example, the highest alpha peaks of the alpha epochs were extracted and averaged [9]. However, brainwave activity is rhythmic or repetitive neural activity that fluctuates in wave-like patterns and depends on the frequency of these patterns. It may not have typical electrophysiological responses or exhibit specific patterns to a stimulus [1, 10, 11]. In general, brainwave activity may be indicated by increases or decreases in frequency and amplitude or show a temporary interruption, which is referred to as phase resetting [12]. Phase resetting can cause rhythms with different starting polarities (positive or negative). Based on these concepts, we wondered whether different brain regions produced or generated different patterns of training-induced alpha activity. Averaging different patterns of phase-locked training-induced alpha activity and then conducting source localization separately should answer the question. In addition, an endogenous or exogenous event can also result in frequency-specific changes to ongoing EEG oscillations that are non-phase-locked to the stimulus and thus cannot be extracted by trial averaging [13]. Spectral analysis techniques, such as time-frequency analysis, are applied to event-related trials to quantify changes. In such analyses, the dynamics of the power of frequency-specific oscillations are quantified, and these spatiotemporal dynamics are examined as they relate to brain processes. Therefore, in addition to phase-locking, spectral analysis could provide another aspect to investigate the generation of training-induced alpha activity.

Considered together, separate analyses of phase-locked and non-phase-locked signals might reveal specific insights into the generation mechanism of training-induced alpha activity. In the present study, we provide four averaging methods for training-induced alpha activity, including phase-locked and non-phase-locked information, to investigate the sources of

training-induced alpha activity through dipole source localization. We hypothesized that the dipole source localization of training-induced alpha activity would be clustered in particular brain regions.

## Materials and methods

### Participants

In this study, the participants were recruited via social media, e.g., Facebook and Instagram, included 63 healthy students studying at National Cheng Kung University from May 2016 to March 2018 using convenience sampling, and they were randomly assigned to two age- and sex-matched groups. The inclusion criteria were Taiwanese nationality, 20–30 years old, right-handed and not having participated in an NFT study in the past. The exclusion criteria included a history of mental or neurological disorders and potential pregnancy. Thirty-five participants (20 women, 24.03 ± 2.60 years old) were allocated to the alpha group (Alpha) and received alpha amplitude feedback, and twenty-eight participants (16 women, 23.68 ± 2.60 years old) were assigned to the control group (Ctrl) and received amplitude feedback in random frequency bands (described below). There was no significant difference in the age ($t = 0.532$, $p = 0.597$), education level ($t = 0.532$, $p = 0.597$), and sex ($\chi^2 = 0$, $p = 1.000$) of the participants in the two groups (Table 1). The experimental procedure was reviewed and approved by the Institutional Review Board of the National Cheng Kung University Hospital (NCKUH-IRB). Informed consent was provided and signed by all participants before the experiment.

### Experimental procedure

The experimental procedure contained a training phase and whole-head EEG recording. A training phase was conducted for all participants, and whole-head EEG recording was conducted only for the "responder" after a training phase. During the training phase, twelve sessions were carried out within 4 weeks (3 sessions per week). Each session contained a 2-min baseline block followed by 6 6-min training blocks with a 1-min inter-block break each. Before the experiment, a trainer was trained to be familiar with the training protocol, ensuring that the trainer could provide instructions and strategies to each participant consistently during NFT. Before NFT, the trainer provided constructive strategies for successful training of alpha activity to each participant, for example, pleasant or relaxing situations such as reading or wandering. The strategies were reported by participants in our previous study [1]. In a baseline block, participants were asked to keep their eyes opened and to not engage in any training event from the training block. During the training block, a digital camera was used to rule out the effects of possible behavioral artifacts, for example, falling asleep/drowsiness, paying less attention during training, or an inadequate strategy involving eye closure or body movement. In the resting period, the researchers used information on the cumulative waveform, e.g., the timestamps of high EEG amplitudes, to help participants recall what kind of strategy they used to achieve a high amplitude. All of the experiments were performed in our laboratory.

**Table 1. Demographics of the Alpha and Ctrl groups.**

|  | Alpha group (N = 35) | Ctrl group (N = 28) | $p$ |
|---|---|---|---|
| Sex (male/female) | 15/20 | 12/16 | 1.000 |
| Age (years) | 24.03 ± 2.60 | 23.68 ± 2.60 | 0.597 |
| Education (years) | 24.03 ± 2.60 | 23.68 ± 2.60 | 0.597 |

## Neurofeedback training and processing

The EEGs of six active Ag/AgCl electrodes located 2.5 cm anterior and posterior to C3, Cz and C4 (C3a, C3p, Cza, Czp, C4a and C4p) were recorded and converted into a bipolar montage by calculating the difference for the electrodes of interest (C3 = C3a-C3p, Cz = Cza-Czp, C4 = C4a-C4p) [14]. The benefit of the bipolar recording was to reduce the possible artifacts of motion or eye blinks [15]. The ground electrode was located at the right mastoid (A2). A homemade multichannel amplifier with a bandpass filter of 0.3–80 Hz was used [16]. Three bipolar EEGs were digitized by an analog-to-digital converter (USB6009, National Instruments, TX) with a 500-Hz sampling rate. All EEG data acquisition and feedback processing were performed in the LabVIEW environment (National Instruments, TX). All raw EEG data were saved for advanced offline processes.

During the training block, the fast Fourier transform (FFT) algorithm with a Hamming window on a second-by-second basis was used to calculate the amplitude spectra of 3 bipolar EEGs. Feedback was given by averaging the amplitude spectra of the 3 bipolar EEGs and updating each second. Feedback information was amplitudes of alpha waves (8-12-Hz) and a particular 4-Hz bandwidth for the Alpha group and Ctrl group, respectively. The 4-Hz bandwidth was randomly selected second-by-second from the range of 7–20 Hz. Thus, the Ctrl group received various kinds of 4-Hz amplitudes during each training block [17].

Two pieces of feedback information were extracted: the instantaneous amplitude of a 1-s averaged amplitude and the cumulative waveform of all 1-s averaged amplitudes. The instantaneous amplitude of a particular frequency band was presented by means of a horizontal bar. If an amplitude increased, the bar moved to the right. Otherwise, the bar moved to the left if the amplitude decreased. Participants were encouraged to move the bar to the rightmost position and hold it there as long as possible. During the resting period between two consecutive training blocks, the researchers used information on the cumulative waveform, e.g., the timestamps of high amplitudes, to help participants recall what kind of strategy they used to achieve a high amplitude.

For both the baseline and training blocks, the FFT algorithm with a Hamming window was used to calculate the alpha amplitude of the 1-s EEG. To verify the training effect of an NFT, the mean relative alpha amplitude was used. The relative alpha amplitude of each 1-s EEG was defined by the alpha amplitude divided by the average alpha amplitude of all 1-s baseline EEGs, as shown below.

$$Relative\ alpha\ amplitude = \frac{Alpha\ amplitude}{Baseline\ alpha\ amplitude}$$

Thus, the mean relative alpha amplitude of a session was calculated by averaging the relative alpha amplitude of all 1-s EEGs.

The present study further defined the well-trained "responder" from the Alpha group to characterize the topographic map and dipole source localization of training-induced alpha activity. The "responder" was identified as a participant exhibiting significantly higher mean relative alpha amplitude per block for the 12th session compared with that for the 1st session (modified from [2]).

## Whole-head EEG and processing

To characterize the topographic map and dipole source localization of training-induced alpha activity, a whole-head EEG was recorded for the "responder" after NFT. The EEG was recorded from 32 Ag/AgCl electrodes placed in a Neuroscan™ Quik-Cap according to the International 10–20 system. All of the electrode positions were measured using a 3D digitizer

(Fastrak, Polhemus, USA). Reference electrodes were attached to the bilateral mastoid processes ((A1+A2)/2). The signals were amplified by a NuAmps amplifier with a 1000-Hz sampling rate and acquired by Scan 4.3 software (Neuroscan, Inc.). The impedance of each electrode was kept below 5 kΩ.

The experimental procedure of the whole-head EEG recording contained 5 runs. Each run interleaved four fixation and four alpha blocks (50 s each). In a fixation block, the "responder" was asked to keep his or her eyes open and not to engage in any training event. In an alpha block, the "responder" was asked to produce training-induced alpha activity. There was a 5-min rest period between two consecutive runs.

The raw data were bandpass filtered between 0.5 and 30 Hz. The alpha amplitude of the 1-s EEG was calculated using the FFT algorithm with a Hamming window for both the fixation and alpha blocks. When the 1-s EEG alpha amplitude of the alpha blocks was higher than the 1.5-fold averaged alpha amplitude of all of the 1-s fixation EEGs, the 1-s EEG segment (epoch) was considered a successful alpha event. The successful alpha events contaminated with blinks, eye-movement artifacts ($> 65$ μV) or remarkable muscle activity were removed. After that, all of the remaining successful alpha events were further processed through four different averaging methods.

The first method was the max peak average (MPA) method. The time points of the highest voltage positive peak of the successful alpha events were identified and defined as trigger points. Based on the time points, 1.2-s alpha epochs centered on the highest positive peak of the artifact-free data were extracted and averaged [9]. The second and third methods were the positive average (PA) and negative average (NA), respectively. The time points of the start of alpha activity from the successful alpha events were identified and defined as trigger points. Based on the time points, 1.2-s alpha epochs with a time window from 0.2 s before the time points to 1 s after the time points were extracted from the artifact-free data. If the first peak voltage in an alpha epoch was positive or negative, it was categorized as a positive alpha epoch or a negative alpha epoch, respectively. Then, the categorized alpha epochs were averaged separately. The fourth method was the event-related spectral perturbation average (ERSPA). All of the categorized alpha epochs were analyzed by ERSP technology. The ERSP technology measured amplitude spectra at each time point. The amplitude spectra of each categorized alpha epoch were averaged over time. An inverse FFT was applied to the averaged ERSP to generate a time series of averaged alpha epoch. Finally, averaged alpha epochs with a time window from 0.5 s before the trigger point to 0.5 s after the trigger point for the MPA method and with a time window from the trigger point to 1 s after the trigger point for the other three methods were used to produce a topographic map and to conduct dipole source localization.

Furthermore, to evaluate the alpha-dominant region, the alpha amplitude of the averaged alpha epoch of the electrodes close to frontal (FP1, FP2, F3, FZ, F4 FC3, FCZ and FC4), parietal (CP3, CPZ, CP4, P3, PZ and P4), occipital (O3, OZ and O4), frontotemporal (F7, F8, FT7 and FT8), and parietotemporal (TP7, TP8, T5 and T6) regions were averaged separately.

## Dipole source localization

Dipole source localization was conducted using Curry 7 (NeuroScan, Inc., USA) software. The spatiotemporal source model was used to model the sources of signals as equivalent current dipoles [18, 19]. In the present study, independent component analysis (ICA) was used to define the numbers of dipoles, and dipole source localization was performed for each "responder". ICA source reconstruction was conducted to compute a fixed dipole source for each independent component [20]. The averaged alpha epoch was applied in ICA before dipole source localization. ICA components were estimated, and we calculated their signal-to-

noise ratio (SNR). The number of dipoles used for dipole source localization depended on the number of valid components (SNR > 1) [21]. A regional dipole model was used to automatically fit the averaged alpha epoch. A standardized boundary element model (BEM) was used as a head model derived from an averaged T1-weighted magnetic resonance image (MRI) dataset, which is available from the Montreal Neurological Institute (MNI, www.mni.mcgill.ca). The measured electrodes were transformed to the BEM coordinate system. The standardized BEM was chosen because it is less prone to spatial errors than other head models (e.g., the spherical conductor model). The standardized BEM improves the location accuracy of bioelectric source reconstruction results by approximating the volume conductor properties of realistically shaped compartments of isotropic and homogeneous conductivities [22]. The skin, skull, and brain compartments are triangulated using mean triangle side lengths (node distances) of 8.3, 6.1, and 4.4 mm. The residual variance (RV) means the percentage of data that cannot be explained by the model. Using the unconstrained dipole solution as a benchmark, a criterion for the RV of <10% was accepted [23].

## Statistical analyses

Demographic characteristics of the two groups were assessed by chi-squared tests and Student's t-tests. To evaluate the alpha NFT course and spectra, two-factor mixed analysis of variance (ANOVA) followed by Bonferroni post hoc testing was conducted on the mean relative alpha amplitude and amplitudes of the delta (1–3 Hz), theta (4–7 Hz), alpha (8–12 Hz), and beta (13–30 Hz) bands. Linear regression analysis was used to examine the relationship between the mean relative alpha amplitude and sessions. To evaluate the alpha-dominant region, Student's t-test was used to compare the difference among the alpha amplitudes of frontal, parietal, occipital, frontotemporal and parietotemporal regions.

The chi-squared test was further used to examine the percentage of participants in each localized brain region compared with zero. The percentage agreement and Cohen's kappa coefficient (κ) were calculated to measure the interrater agreement on dipole source localization between each method. The kappa value generally ranges from 0 to 1.0, where different subranges indicate different levels of agreement (< 0, poor agreement; 0.00–0.20, slight agreement; 0.21–0.40, fair agreement; 0.41–0.60, moderate agreement; 0.61–0.80, substantial agreement; 0.81–1.00, almost perfect agreement) [24]. To assess the agreement on the brain regions of source localizations among the methods, agreement plots were used with 18 x 18 matrices that matched each brain region for each participant. All statistical analyses were performed using SPSS version 17.0. The data are expressed as means ± standard errors of the means. The two-tailed significance level was set to 0.05.

## Results

### EEG data for NFT

To ascertain the NFT effect, the mean relative alpha amplitude across the 12 sessions was used to reflect the dynamic changes in the alpha amplitude. The mean relative alpha amplitude progressively increased throughout the 12 sessions in the Alpha group but not in the Ctrl group (Fig 1). ANOVA revealed significant main effects of group ($F_{1,61}$ = 16.382, $p < 0.001$), session ($F_{11,671}$ = 5.726, $p < 0.001$), and their interaction ($F_{11,671}$ = 4.642, $p < 0.001$). The mean relative alpha amplitude showed no significant difference in the Ctrl group throughout NFT. In contrast, the mean relative alpha amplitude of the 5th-12th sessions in the Alpha group significantly higher than that of the 1st session, and they also significantly higher than that of the Control group. In addition, a linear increase was observed in the Alpha group ($R^2$ = 0.103, $p < 0.001$).

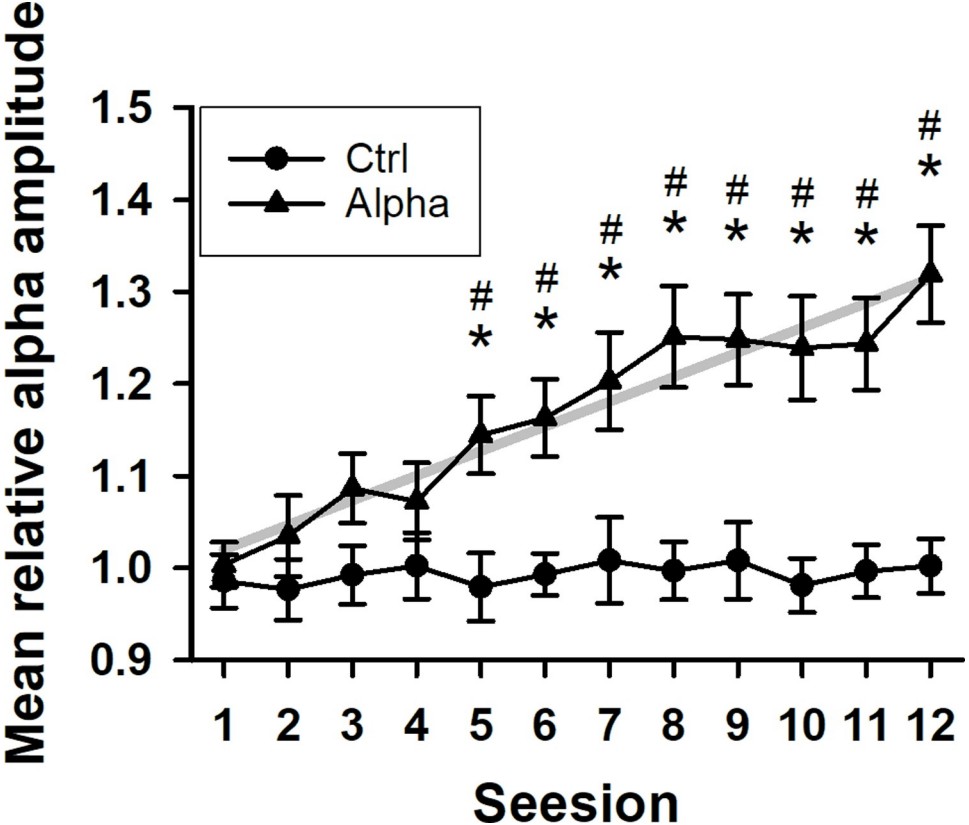

**Fig 1. Mean relative alpha amplitude changes throughout the 12 sessions in two groups.** The gray line is the linear regression line and indicates a progressive linear increase in the Alpha group. $^*$ $p < 0.05$ vs. the 1st session. $^\#$ $p < 0.05$ vs. the Ctrl group.

To further examine changes in the amplitude spectra of both groups (Fig 2), the amplitude spectra between the 1st and 12th sessions were analyzed. The amplitude spectra of the Ctrl group showed no significant difference between the 1st and 12th sessions. For the delta, theta and beta bands, ANOVA revealed no significant main effect of group (delta, $F_{1,61} = 3.281$, $p = 0.075$; theta, $F_{1,61} = 2.465$, $p = 0.122$; beta, $F_{1,61} = 0.644$, $p = 0.425$), session (delta, $F_{1,61} = 1.427$, $p = 0.237$; theta, $F_{1,61} = 3.228$, $p = 0.077$; beta, $F_{1,61} = 1.166$, $p = 0.285$), and their interaction (delta, $F_{1,61} = 0.002$, $p = 0.966$; theta, $F_{1,61} = 0.230$, $p = 0.634$; beta, $F_{1,61} = 0.488$, $p = 0.488$). In contrast, for the alpha band, significant main effects of group ($F_{1,61} = 4.535$, $p = 0.037$), session ($F_{1,61} = 20.530$, $p < 0.001$), and their interaction ($F_{1,61} = 4.466$, $p = 0.039$). The alpha band showed a significant difference between the two groups in the 12th session, but not in the 1st session. A significant difference between the 1st and 12th sessions in the Alpha group was also observed. In particular, amplitudes of the 9-, 10-, and 11-Hz waves in the 12th session were significantly higher than those in the 1st session.

Twenty-nine participants (83%) were identified as "responders" in this study. Therefore, six participants in the Alpha group who were not well trained were excluded from whole-head EEG data, EEG mapping and dipole source localization analyses.

## Whole-head EEG data

The alpha amplitudes of the alpha (13.02 ± 1.01 μV) and fixation (10.01 ± 0.71 μV) blocks were significant different (t = 4.822, $p < 0.001$). Besides, a significant difference between the

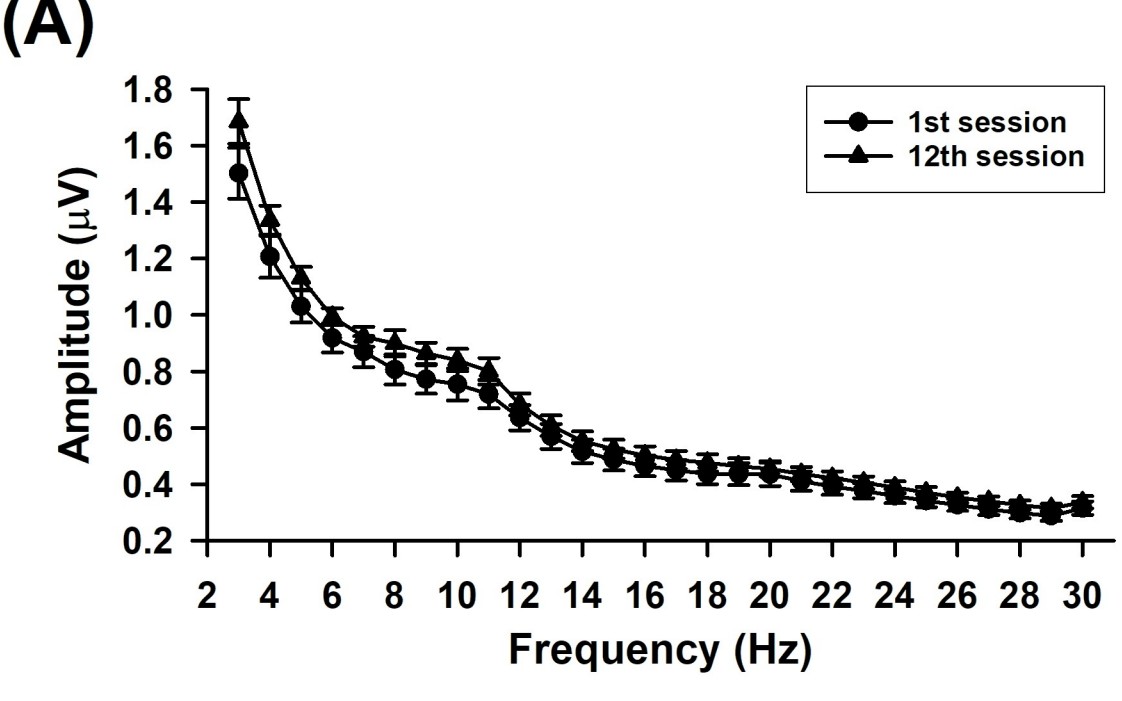

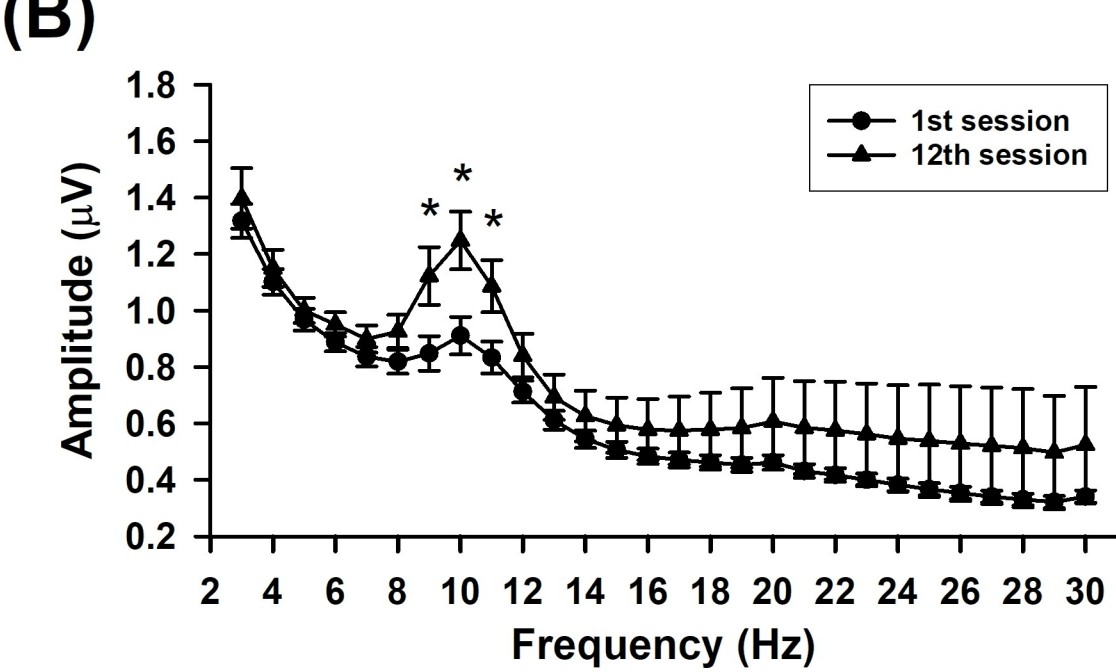

**Fig 2. Amplitude spectra of the 1st and 12th sessions in the two groups.** The (A) Ctrl group and (B) Alpha group. * $p < 0.05$ vs. the 1st session.

successful alpha events of the alpha (324.24 ± 22.27, range: 224–606) and fixation blocks (55.03 ± 3.97, range: 11–89) was also observed (t = 12.53, $p < 0.001$). These data indicated that "responders" had learned to successfully induce alpha activity in the alpha block. The successful alpha events were further processed through four different averaging methods. Finally,

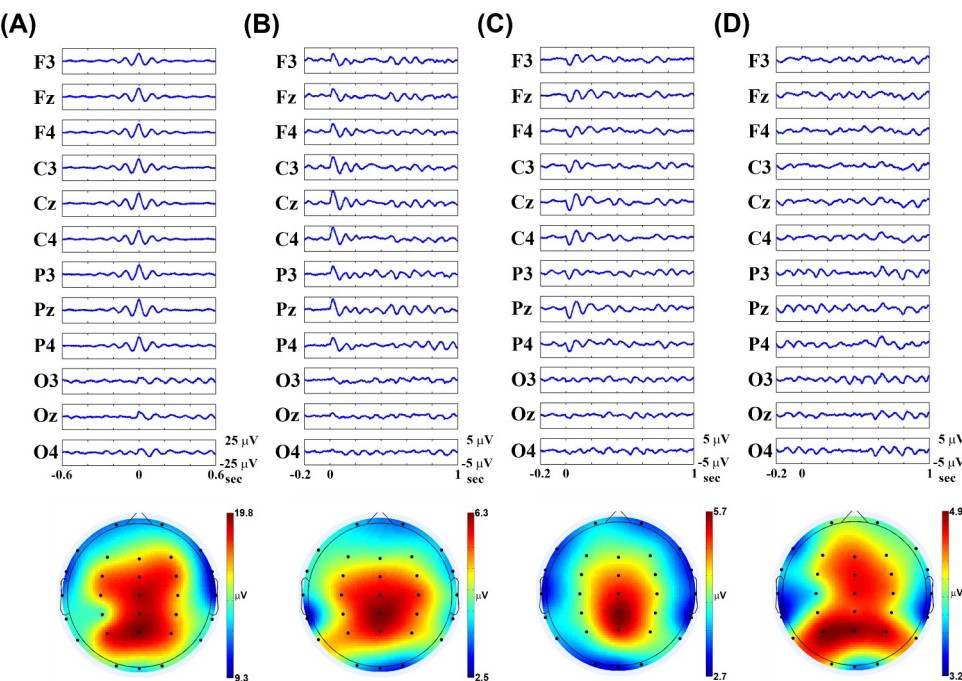

**Fig 3. The grand-averaged alpha epochs (top row) with corresponding topographic maps (bottom row) for the four methods.** (A) max peak average (MPA), (B) positive average (PA), (C) negative average (NA) and (D) event-related spectral perturbation average (ERSPA) methods.

alpha epochs of the MPA and ERSPA (324.24 ± 22.27, range: 224–606), PA (186.03 ± 14.27, range: 66–358) and NA (138.21 ± 10.18, rang: 61–285) methods were identified and averaged.

## EEG mapping of training-induced alpha activity

To characterize the topographic map of training-induced alpha activity, the grand-averaged alpha epochs (top row) with corresponding topographic maps (bottom row) for the four methods are shown in Fig 3. For the MPA method, the highest peak in the middle of the alpha epoch and a flat period of both sides were observed. Significant differences among the alpha amplitude of parietal and frontal (t = 2.137, $p$ = 0.041), parietal and occipital (t = 6.558, $p < 0.001$), parietal and frontotemporal (t = 6.583, $p < 0.001$), parietal and parietotemporal (t = 7.335, $p < 0.001$), frontal and frontotemporal (t = 15.239, $p < 0.001$), frontal and parietotemporal (t = 2.315, $p$ = 0.028), occipital and frontotemporal (t = 3.680, $p$ = 0.001), and parietotemporal and frontotemporal (t = 3.640, $p$ = 0.001) regions were observed. For the PA method, the largest positive peak followed the start of the alpha epoch (trigger point). Significant differences among the alpha amplitude of parietal and frontal (t = 2.340, $p$ = 0.027), parietal and occipital (t = 5.699, $p < 0.001$), parietal and frontotemporal (t = 4.521, $p < 0.001$), parietal and parietotemporal (t = 4.555, $p < 0.001$), frontal and occipital (t = 1.793, $p$ = 0.007), frontal and frontotemporal (t = 2.778, $p$ = 0.010), occipital and frontotemporal (t = -0.784, $p$ = 0.035), and occipital and parietotemporal (t = -1.055, $p$ = 0.001) regions were observed. For the NA methods, the negative peak followed the start of the alpha epoch (trigger point). Significant differences among the alpha amplitude of parietal and frontal (t = 1.356, $p$ = 0.020), parietal and occipital (t = 4.518, $p < 0.001$), parietal and frontotemporal (t = 4.424, $p < 0.001$), parietal and parietotemporal (t = 5.217, $p < 0.001$), frontal and occipital (t = 1.627, $p$ = 0.040), frontal and frontotemporal (t = 5.969, $p < 0.001$), occipital and parietotemporal (t = -0.824, $p$ = 0.032),

and occipital and frontotemporal (t = 1.003, $p$ = 0.042) regions were observed. Before the trigger point, a flat period was observed in the PA and NA methods. For the ERSPA method, alpha activity was obviously observed after inverse FFT implementation. Significant differences among the alpha amplitude of parietal and frontal (t = 0.714, $p$ = 0.045), parietal and occipital (t = 1.077, $p$ = 0.003), parietal and frontotemporal (t = 1.751, $p$ = 0.002), parietal and parietotemporal (t = 2.568, $p$ = 0.002), frontal and occipital (t = 0.622, $p$ = 0.042), frontal and frontotemporal (t = 1.332, $p$ = 0.049), occipital and parietotemporal (t = 0.299, $p$ = 0.026), and parietotemporal and frontotemporal (t = 0.195, $p$ = 0.028) regions were observed. These data indicated that training-induced alpha activity of parietal region was significantly higher than other regions.

## Dipole source localization

To identify the sources of training-induced alpha activity, we proposed four different averaging methods to estimate the sources. Four "responders" were excluded due to the RV were greater than 10%. In the present study, the dipole clusters of training-induced alpha activity for the four methods showed similar distribution in the brain, primarily located in the precuneus, posterior cingulate cortex (PCC) and middle temporal gyrus (Fig 4). The dipole source localization revealed that the RV for dipole estimation with the MPA method was 4.15 ± 0.50%, which was less than that with the PA (6.91 ± 0.43%, t = -5.516, $p$ < 0.001), NA (7.34 ± 0.33%, t = -7.318, $p$ < 0.001) and ERSPA (6.74 ± 0.34%, t = -4.484, $p$ < 0.001) methods. No significant difference was found between RVs of the PA, NA and ERSPA methods. All the RVs were less than 10%, revealing that dipole estimation with the four methods showed a good fit to training-induced alpha activity. The detailed coordinates and distribution of the dipoles are summarized in Table 2. For the MPA method, the percentages of "responders" whose dipoles were clustered in the precuneus, PCC and middle temporal areas were 84% ($p$ < 0.001), 68% ($p$ < 0.001) and 40% ($p$ < 0.001) significantly different from zero, respectively. For the PA method, the percentages of "responders" whose dipoles were clustered in the

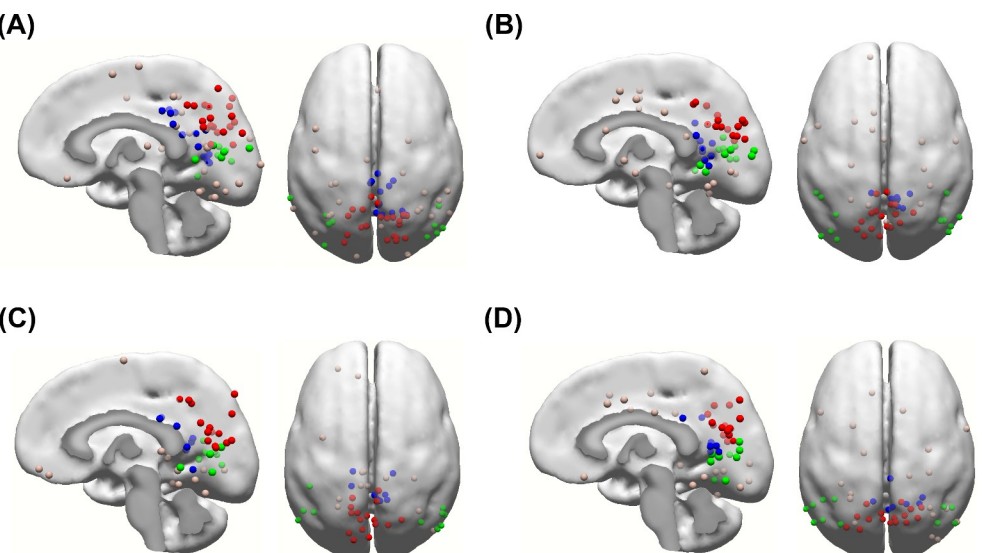

**Fig 4. Distribution of equivalent current dipoles of training-induced alpha activity for the four methods.** The (A) max peak average (MPA), (B) positive average (PA), (C) negative average (NA) and (D) event-related spectral perturbation average (ERSPA) methods. Note that the red dots are located close to the precuneus, the blue dots are located close to the posterior cingulate cortex (PCC), and the green dots are located close to the middle temporal gyrus.

**Table 2. Coordinates of training-induced alpha sources for the four averaging methods.**

| Area | | MPA | | | | PA | | | | NA | | | | ERSPA | | | |
|---|---|---|---|---|---|---|---|---|---|---|---|---|---|---|---|---|---|
| | | MNI coordinates (mm) | | | N (%) | MNI coordinates (mm) | | | N (%) | MNI coordinates (mm) | | | N (%) | MNI coordinates (mm) | | | N (%) |
| | | x | y | z | | x | y | z | | x | y | z | | x | y | z | |
| Precuneus | R | 15.6 | -63.5 | 38.3 | 21 (84%) | 9.0 | -56.7 | 34.3 | 18 (72%) | 6.1 | -60.1 | 41.1 | 16 (64%) | 15.0 | -60.0 | 32.6 | 16 (64%) |
| | L | -12.8 | -59.6 | 39.2 | | -12.3 | -64.1 | 32.2 | | -8.1 | -62.8 | 26.9 | | -14.4 | -62.3 | 33.3 | |
| PCC | R | 11.6 | -43.2 | 23.8 | 17 (68%) | 12.0 | -51.9 | 15.0 | 9 (36%) | 10.8 | -41.6 | 19.8 | 10 (40%) | 10.9 | -50.9 | 21.2 | 8 (32%) |
| | L | -4.9 | -23.8 | 33.3 | | -9.0 | -43.7 | 22.3 | | -8.3 | -22.1 | 24.9 | | -12.6 | -32.6 | 23.1 | |
| Middle temporal | R | 49.4 | -71.3 | 13.3 | 10 (40%) | 51.3 | -67.5 | 14.9 | 12 (48%) | 53.8 | -67.8 | 13.4 | 10 (40%) | 49.6 | -65.3 | 12.6 | 11 (44%) |
| | L | -41.3 | -66.4 | 16.4 | | -44.5 | -65.4 | 10.3 | | -50.3 | -59.7 | 8.0 | | -49.9 | -62.9 | 10.2 | |
| Superior temporal | R | 59.0 | -58.9 | 21.5 | 2 (8%) | - | - | - | - | - | - | - | - | - | - | - | - |
| | L | -64.5 | -56.1 | 17.4 | | - | - | - | | - | - | - | | - | - | - | |
| Inferior temporal | R | - | - | - | 1 (4%) | - | - | - | - | - | - | - | - | - | - | - | - |
| | L | -61.5 | -28.5 | -24.0 | | - | - | - | | - | - | - | | - | - | - | |
| Middle frontal | R | 25.6 | 2.3 | 42.2 | 3 (12%) | 28.5 | 0.4 | 43.8 | 3 (12%) | - | - | - | 1 (4%) | 31.8 | -6.2 | 44.8 | 3 (12%) |
| | L | -42.6 | 4.9 | 56.7 | | -48.4 | 11.6 | 40.0 | | -34.9 | 5.3 | 61.8 | | -36.3 | 14.3 | 31.8 | |
| Medial frontal | R | 3.9 | 31.2 | -21.3 | 1 (4%) | -9.9 | 57.6 | 11.6 | 1 (4%) | - | - | - | 2 (8%) | - | - | - | 1 (4%) |
| | L | - | - | - | | - | - | - | | -17.3 | 43.5 | -21.0 | | -10.1 | 26.5 | -21.6 | |
| Inferior parietal | R | 44.2 | -42.7 | 33.5 | 3 (12%) | - | - | - | - | - | - | - | - | - | - | - | - |
| | L | -32.2 | -34.5 | 27.8 | | - | - | - | | - | - | - | | - | - | - | |
| Lingual | R | 6.6 | -78.2 | -8.9 | 1 (4%) | 3.3 | -74.1 | 2.4 | 3 (12%) | - | - | - | - | - | - | - | 1 (4%) |
| | L | - | - | - | | -11.4 | -61.4 | -7.4 | | - | - | - | | -24.0 | -77.7 | -7.4 | |
| Cuneus | R | 26.4 | -89.2 | 31.0 | 2 (8%) | - | - | - | - | - | - | - | - | - | - | - | - |
| | L | -13.0 | -96.3 | 14.1 | | - | - | - | | - | - | - | | - | - | - | |
| Fusiform | R | 43.3 | -67.8 | -14.2 | 4 (16%) | 31.0 | -62.4 | -15.8 | 2 (8%) | 39.5 | -60.4 | -16.2 | 3 (12%) | 50.7 | -68.3 | -17.6 | 1 (4%) |
| | L | -30.5 | -63.7 | -12.7 | | -25.6 | -57.4 | -10.0 | | -27.2 | -53.5 | -17.1 | | - | - | - | |
| Thalamus | R | 23.1 | -29.1 | 11.7 | 2 (8%) | - | - | - | - | 12.5 | -36.9 | 4.3 | 3 (12%) | 0.1 | -2.0 | 4.5 | 1 (4%) |
| | L | -4.8 | -18.8 | 11.4 | | - | - | - | | -5.5 | -31.2 | 8.1 | | - | - | - | |
| Precentral | R | - | - | - | - | 54.2 | -0.8 | 34.6 | 2 (8%) | - | - | - | - | 65.6 | 6.7 | 31.5 | 1 (4%) |
| | L | - | - | - | | -59.0 | -0.2 | 36.5 | | - | - | - | | - | - | - | |
| Postcentral | R | 54.2 | -34.0 | 51.7 | 2 (8%) | - | - | - | - | - | - | - | - | - | - | - | 1 (44%) |
| | L | -28.3 | -29.2 | 46.7 | | - | - | - | | - | - | - | | -25.7 | -39.0 | 63.7 | |
| Middle occipital | R | 35.4 | -62.4 | 1.8 | 3 (12%) | - | - | - | - | 44.8 | -72.3 | 2.4 | 2 (8%) | 38.9 | -72.6 | 3.0 | 3 (12%) |
| | L | -41.0 | -74.9 | -2.8 | | - | - | - | | - | - | - | | - | - | - | |
| Para-hippocampal | R | - | - | - | 1(4%) | 33.4 | -33.3 | 16.0 | 1 (4%) | 33.4 | -44.4 | -12.9 | 4 (16%) | 19.6 | -36.8 | -1.4 | 3 (12%) |
| | L | -16.0 | -28.7 | -14.5 | | - | - | - | | -11.3 | -49.7 | -6.9 | | -31.0 | -49.3 | -8.8 | |
| Insula | R | - | - | - | - | 36.2 | -38.5 | 20.9 | 1 (4%) | 33.7 | -33.0 | 22.2 | 1 (4%) | - | - | - | 2 (8%) |
| | L | - | - | - | | - | - | - | | - | - | - | | -35.0 | -21.1 | 21.2 | |
| ACC | R | - | - | - | 1 (4%) | - | - | - | - | - | - | - | - | 13.0 | 21.6 | 37.2 | 1 (4%) |
| | L | -3.3 | 23.1 | -5.7 | | - | - | - | | - | - | - | | - | - | - | |

Values of coordinates are means.

N, number of "responders". MPA, max peak average; PA, positive average; NA, negative average; ERSPA, event-related spectral perturbation average. PCC, posterior cingulate cortex; ACC, anterior cingulate cortex.

precuneus, PCC and middle temporal areas were 72% ($p < 0.001$), 36% ($p = 0.001$) and 48% ($p < 0.001$) significantly different from zero, respectively. For the NA method, the percentages of "responders" whose dipoles were clustered in the precuneus, PCC and middle temporal areas were 64% ($p < 0.001$), 40% ($p = 0.005$) and 40% ($p < 0.001$) significantly different from

**Table 3. Interrater agreement on dipole source localization between each method.**

|  | MPA vs. PA | MPA vs. NA | MPA vs. ERSPA | PA vs. NA | PA vs. ERSPA | NA vs. ERSPA |
|---|---|---|---|---|---|---|
| **% agreement** | 79.2 | 69.6 | 83.6 | 82.8 | 86.7 | 84.4 |
| **Cohen's kappa (κ)** | 0.75* | 0.62* | 0.80* | 0.79* | 0.84* | 0.81* |

MPA, max peak average; PA, positive average; NA, negative average; ERSPA, event-related spectral perturbation average.

* $p < 0.05$, significantly different from zero.

zero, respectively. For the ERSPA method, the percentages of "responders" whose dipoles were clustered in the precuneus, PCC and middle temporal areas were 64% ($p < 0.001$), 32% ($p = 0.002$) and 44% ($p < 0.001$) significantly different from zero, respectively. The percentage of "responder" whose dipoles were located in the other regions was $< 20\%$ for each method and showed no significant different from zero.

## Agreement

The percentage agreement and Cohen's kappa for each averaging method are shown in Table 3. Substantial agreement between the MPA and PA methods (79.2%, κ = 0.75) and MPA and NA methods (69.6%, κ = 0.62) was observed for dipole source localization. Almost perfect agreement between the MPA and ERSPA methods (83.6, κ = 0.80), the PA and NA methods (82.8%, κ = 0.79), the PA and ERSPA methods (86.7%, κ = 0.84), and the NA and ERSPA methods (84.4%, κ = 0.81) were observed after dipole source localization. All Cohen's kappa coefficients (κ) were significantly different from zero ($p < 0.05$). Agreement plots were used to present the dipole source localization agreement between each method. Good agreement was observed in the precuneus between the MPA and PA methods (Fig 5A), as well as between the MPA and ERSPA methods (Fig 5C). Between the MPA and NA methods, good agreement was observed in the precuneus and PCC (Fig 5B). Good agreement was observed in the precuneus and middle temporal gyrus between the PA and NA methods (Fig 5D), as well as between the PA and ERSPA methods (Fig 5E). Between the NA and ERSPA methods, good agreement was observed in the precuneus, middle temporal gyrus and PCC (Fig 5F). The present results revealed that good agreement was observed mainly in the precuneus, PCC and middle temporal gyrus.

## Discussion

The present study aimed to identify dipoles of alpha activity after NFT through four temporal/spectral analytic techniques, i.e., MPA, PA, NA and ERSPA methods. Before dipole source analysis, we confirmed that the Alpha group showed significant successful alpha activity training in terms of linear changes in the mean relative alpha amplitude throughout the 12 sessions. Alpha NFT significantly affected only the frequency spectrum in the trained alpha brand and not in the other bands. Through the MPA, PA, NA and ERSPA methods, the topographic maps of training-induced alpha activity were mainly over parietal area. Similar results among the four methods showed that the dipole clusters of training-induced alpha activity were located in the precuneus, PCC and middle temporal gyrus. The RV (goodness of fit) for dipole estimation of the MPA method was significantly smaller than that of the other methods. Good agreement between each method was also observed mainly in three brain regions. Our findings indicates that the precuneus, PCC and middle temporal gyrus seem to play important roles in enhancing training-induced alpha activity.

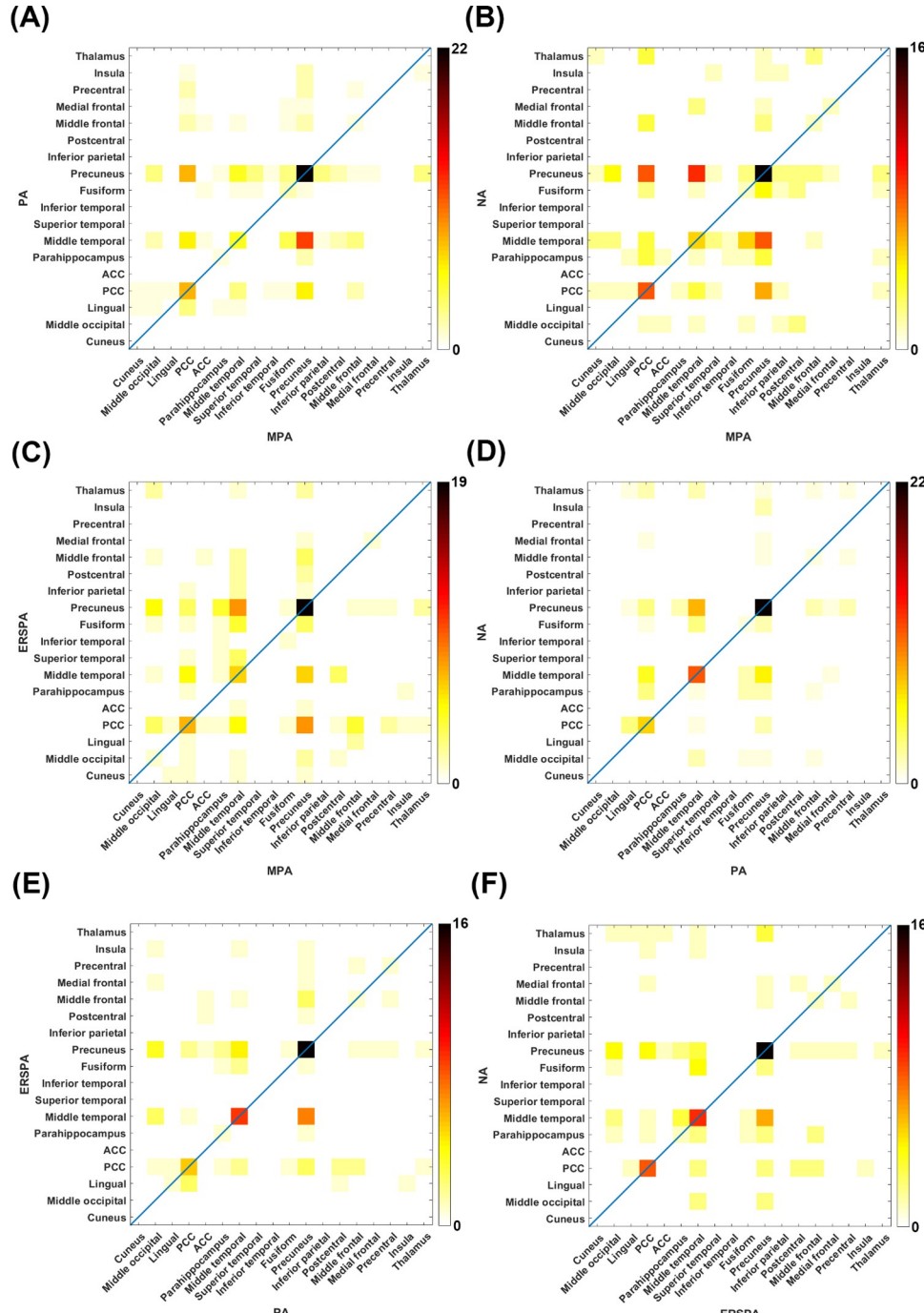

**Fig 5. Agreement plots for dipole source localization between each method.** The (A) MPA and PA, (B) MPA and NA, (C) MPA and ERSPA, (D) PA and NA, (E) PA and ERSPA, and (F) NA and ERSPA methods. Note that good agreement is observed mainly in the precuneus, PCC and middle temporal gyrus. MPA, max peak average; PA, positive average; NA, negative average; ERSPA, event-related spectral perturbation average; PCC, posterior cingulate cortex; ACC, anterior cingulate cortex.

A linear increase in the amplitude of the alpha band, but not the other frequency bands, was observed after NFT. Our findings were in line with previous studies confirming the train-ability and independence of alpha NFT [1, 2]. Participants enhanced their alpha activity and

controlled the alpha band independently of the other frequency bands, showing that alpha NFT affected only the trained alpha activity and did not significantly change the other frequency bands. This is remarkable because stable and reliable alpha activity is helpful for localizing dipole sources.

Source analysis is primarily used to estimate the dipole source of grand-averaged ERP components [25–27]. Compared to the average trial, a single trial of brainwaves may have a low SNR resulting in increased error (RV) and underestimation during dipole source localization. For this concept, we averaged the max peak of the enhanced alpha activity and successfully obtained a good RV ($< 5$) after dipole analysis [9]. Additionally, considering the different starting polarities of the alpha peak, the enhanced alpha epochs were separated into two categories, positive and negative, and then averaged. After dipole source analysis, good RVs ($< 10$) were also observed, which is similar to the results of a previous study [28]. Interestingly, the averaged epochs showed a decreased amplitude of fluctuation from the starting time points or time points of the highest alpha peak to the end of epoch. The possible reason may be due to non-phase-locked components of alpha activity. The contribution of the averaged alpha epoch in the time domain mainly comes from the phase-locked component [29]. The non-phase-locked component may cause a different phase change in each alpha epoch, resulting in a decreased amplitude of fluctuation after averaging. Therefore, we further used the ERSPA method to combine both the phase-locked and non-phase-locked components of enhanced alpha activity, which showed an obvious alpha activity and a good RV ($< 10$) after inverse FFT and dipole analysis. We used four different average methods for dipole analysis of training-induced alpha activity and obtained satisfactory RVs. Our findings indicated that four average methods, especially MPA, were suitable for investigating sources of brainwaves.

The RV from the MPA method was significantly smaller than that of the other methods, indicating that a greater percentage of the data can be explained by the MPA method than the other methods. The possible reason may be due to the different number of alpha epochs. The number of epochs for the PA and NA methods were separated from that for the MPA method. For this reason, the number of epochs for the MPA method was larger than that of the PA and NA methods, resulting in a greater SNR to decrease the RV of dipole source reconstruction [8]. Interestingly, although the number of epochs for the MPA and ERSPA methods were the same, the RV of the MPA method was also significantly smaller than that of the ERSPA method. The possible reason may be that the noise component of alpha epochs was retained when calculating the ERSPA, resulting in a lower SNR of the alpha epoch after performing an inverse FFT than that of the MPA.

In general, researchers have made an assumption regarding dipole fitting according to neurophysiological and anatomical knowledge from previous studies. This assumption can help researchers decide how many dipoles can be used for dipole fitting [1, 23, 30]. However, no recent study has provided information about the sources of training-induced alpha activity. In the percent study, ICA source reconstruction computed a fixed dipole source for each independent component [20]. Valid components (SNR $> 1$) helped us to define how many dipoles to use for dipole source localization. Training-induced alpha activity was averaged through four methods, and dipole source localization was further applied. We found that the sources of training-induced alpha activity were mostly clustered in the precuneus, PCC and middle temporal gyrus. Good agreement between each method was also observed mainly in three brain regions. Considered together, our findings provide valuable information for future studies that three dipoles could be used for advanced analysis of training-induced alpha activity, especially the training sites are around the central regions.

Alpha activity has been thought to be associated with cognitive functions. For example, alpha power increases were found during internal attention task [31] and decreases during

demanding arithmetic task [32]. In the present study, participants learned to self-regulate alpha activity by shifting internal (mental imagery) and external attention (feedback) during training. Such shift produces comparable signatures of alpha activity modulations [33]. Alpha activity increases reflect internal attention of mental imagery [34]. In neuroimaging studies, the precuneus is involved in visuospatial attention functions [35]. The PCC regulates the focus of attention and control the balance between internal and external attention [36]. Lesions of the middle temporal gyrus were associated with spatial attention deficits in spatial neglect patients [37]. All of the evidence supports the existence of a range of attention-related influences on activation in the precuneus, PCC and middle temporal gyrus. The amplitude of alpha increases reflects cortical inhibition [38]. Inhibition of these three brain regions could protect reorienting against irrelevant stimulation, which could modulate attention function more sufficiently. Consequently, after training, participants might efficiently modulate internal attention to increase alpha activity. Internal attention involves modulation, selection, and maintenance of internal produced information, such as long-term memory or working memory [39]. Considered together, alpha NFT could potentially modulate these three attention-related regions to improve internal attention function and further affect memory performance. Our findings could provide valuable information to partially support memory improvement after alpha NFT in previous studies.

In the present study, a larger alpha amplitude was observed in the parietal region than in other regions indicated that training-induced alpha activity dominated in the parietal region. The result was similar with the dipoles locations (the PCC and precuneus). Besides, alpha activity also showed slightly higher in the frontal region. Alpha NFT was thought to modulate top-down control networks, resulting in participants learned successfully to self-regulate their alpha activity. An increased frontal alpha amplitude may enable a tight functional coupling between frontal cortical areas, thereby allowing the control of the visual processing [40]. Alpha oscillations of frontal brain areas seem to be important for the top-down control [41, 42]. A recent simultaneous EEG-fMRI study further found that long-range alpha synchrony was intrinsically linked to activity in the frontal-parietal control network [43]. This evidence may support that frontal alpha activity was slightly enhanced by alpha NFT. Another possible reason of increased frontal alpha amplitude may be due to "neural efficiency". NFT could cause the neurons of the frontal region to integrate information over space and time repeatedly, causing "neural efficiency" [44]. Neural efficiency hypothesizes that neural activation is decreased in experts. Well-trained "Responders" as experts had learned to decrease neural activations of frontal region after NFT, resulting in increased frontal alpha activity.

One of the limitations of the present study was low-density EEG. The low-density EEG used in dipole source localization may cause blurring and localization error. However, low-density EEG (<32 electrodes) used in dipole source localization may obtain valuable insight in application with some brain activities, such as epileptic spikes [45]. Although the amplitude of alpha activity may be not greater than that of epileptic activity, we used average method to increase SNR of alpha activity. A good SNR can decrease the estimation error of dipole source localization. Indeed, more electrodes could reduce more distance errors in dipole source analysis [46]. More electrodes, such as 64 electrodes or more, should be tested in future research. The other limitation was absence of individual anatomical images. Individual anatomical image seems toimprove the source localization accuracy. However, a previous study has indicated the similarity in source localization between individual and standardized BEM models [22]. The evidence suggest that standardized BEM models used in this study may remain sufficient.

## Conclusions

To our knowledge, no study has investigated dipoles of training-induced alpha activity. In the present study, four averaging methods were used for dipole estimation of training-induced alpha activity with phase-locked and non-phase-locked information. Dipole clusters of training-induced alpha activity were mostly located in the precuneus, PCC and middle temporal gyrus. Our findings provide valuable evidence that the four averaging methods (especially the MPA method) are suitable for investigating sources of brainwaves in future studies, and three dipoles can be used for dipole source analysis of training-induced alpha activity, especially the training sites are around the central regions.

## Acknowledgments

The authors thank the Mind Research and Imaging Center (MRIC) at National Cheng Kung University for consultation and instrument availability.

## Author Contributions

**Conceptualization:** Jen-Jui Hsueh, Fu-Zen Shaw.

**Data curation:** Jen-Jui Hsueh, Fu-Zen Shaw.

**Formal analysis:** Jen-Jui Hsueh, Yan-Zhou Chen.

**Funding acquisition:** Fu-Zen Shaw.

**Investigation:** Jen-Jui Hsueh, Yan-Zhou Chen.

**Methodology:** Jen-Jui Hsueh.

**Project administration:** Fu-Zen Shaw.

**Resources:** Fu-Zen Shaw.

**Software:** Jen-Jui Hsueh.

**Supervision:** Jia-Jin Chen, Fu-Zen Shaw.

**Validation:** Jia-Jin Chen, Fu-Zen Shaw.

**Visualization:** Jen-Jui Hsueh.

**Writing – original draft:** Jen-Jui Hsueh.

**Writing – review & editing:** Fu-Zen Shaw.

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
