## [Decision Letter · Decision Letter 0]

22 Sep 2020

PONE-D-20-20804

Equivalent current dipole sources of neurofeedback training-induced alpha activity through temporal/spectral analytic techniques.

PLOS ONE

Dear Dr. Shaw,

Thank you for submitting your manuscript to PLOS ONE. After careful consideration, we feel that it has merit but does not fully meet PLOS ONE’s publication criteria as it currently stands. Therefore, we invite you to submit a revised version of the manuscript that addresses the points raised during the review process.

We look forward to receiving your revised manuscript.

Kind regards,

Alice Mado Proverbio

Academic Editor

PLOS ONE

Journal Requirements:

2. In your Methods section, please provide additional information about the participant recruitment method and the demographic details of your participants. Please ensure you have provided sufficient details to replicate the analyses such as: a) the recruitment date range (month and year), b) a description of any inclusion/exclusion criteria that were applied to participant recruitment, c) a table of relevant demographic details, d) a statement as to whether your sample can be considered representative of a larger population, e) a description of how participants were recruited, and f) descriptions of where participants were recruited and where the research took place.

3. Thank you for your ethics statement:

'The experimental procedure was reviewed and approved by a local research ethics committee. Informed consent was provided and signed by all participants before the experiment.'

"The authors thank the following institutions for their research support: Ministry of

Science and Technology, Taiwan (108-2410-H-006-112-MY3 and 108-2634-F-006-

012), and the Mind Research and Imaging Center, National Cheng Kung University,

Taiwan."

Reviewers' comments:

Reviewer's Responses to Questions

**Comments to the Author**

1. Is the manuscript technically sound, and do the data support the conclusions?

Reviewer #1: Partly

Reviewer #2: Partly

Reviewer #3: No

Reviewer #4: Partly

2. Has the statistical analysis been performed appropriately and rigorously? 

Reviewer #1: Yes

Reviewer #2: No

Reviewer #3: I Don't Know

Reviewer #4: Yes

3. Have the authors made all data underlying the findings in their manuscript fully available?

Reviewer #1: No

Reviewer #2: Yes

Reviewer #3: Yes

Reviewer #4: Yes

4. Is the manuscript presented in an intelligible fashion and written in standard English?

Reviewer #1: Yes

Reviewer #2: Yes

Reviewer #3: Yes

Reviewer #4: Yes

5. Review Comments to the Author

Reviewer #1: The manuscript titled “Equivalent current dipole sources of neurofeedback training-induced alpha activity through temporal/spectral analytic techniques” touches upon evergreen subject of EEG neurofeedback. It comes with reasonable number of participants (63) and an interesting attempt to localize sources of observed EEG activity.

Although interesting some major questions need to be answered before final conclusions can be made:

1. The training protocol was based on the alpha band which is highly susceptible to various manipulation (Williams, 1977; Quandt et al. 2012) and the training protocol included instructions for participants. How many trainer did provide those instructions and how authors assured mitigation of trainer effect ?

2. Were the participants personality profiles investigated (using psychological tests such as EPQ-R, NEO-EFI,…) ?

Authors’ results are based only on the “responders” subgroup of the “alpha” – experimental group:

1. How many exactly “responders” were finally identified ?

2. Are there any specific features allowing to distinguish responders vs non-non-responders other than increase in alpha amplitude ?

3. Is there any relations between responders, non-responders and instructions, trainers or personality traits ?

4. What would be result for comparison all “alpha”(experimental) group vs “ctrl” (control) group ?

Authors used bipolar electrode montage very rarely used in this kind of investigations which makes it difficult to compare to similar studies:

1. Why Authors decided to use bipolar montage ?

2. What would be the results recalculated to the average reference ?

3. What was the results of whole head analyses for “non-responders” ?

Source analyses

1. Sources were calculated using Curry software which the procedure is not publicly available, could you provide detailed methodology, please ?

2. From my understanding sources were calculated using ICA. Since ICA will return different clusters for each participants how the results were averaged: calculated individually, finding common clusters and then averaged or they were all subjects were analysed together ?

3. Without individual electrode positions and/or individual T1s localization results will not provide accurate results allowing for anatomical consierations

Other methodological questions:

4. Authors investigated also the delta, theta and beta (13-30 Hz) bands what was the results how they compare to alpha band results (especially the theta band which could possibly drive the alpha one) ?

5. What was exactly calculation of ERSPA ?

Reviewer #2: The authors aimed to explore the neurophysiological sources of alpha increases induced by neurofeedback training. Therefore, they calculated the power increases during 12 sessions of alpha neurofeedback training by using four different analysis techniques, namely the maximum peak average, the positive average, the negative average, and the event-related spectral perturbation average. All analysis methods showed a high agreement on localized dipole sources in the precuneus, posterior cingulate cortex, and middle temporal gyrus. The authors conclude that all measures are suitable for investigating dipole sources during neurofeedback training.

The research question itself is an interesting one and worth to be investigated. However, it is unclear how the authors derived the research question and why the described methods were chosen in order to answer it. The method section lacks important descriptions and it is not comprehensible why and on the basis of which criteria the authors ended up with a source localization analysis with 14 participants despite 63 that were tested. It is questionable why four analysis techniques were compared, considering that the research question focused on exploration of the source regions of alpha activity, and it is not reported whether source localization would have been significant using solely one of these methods. Furthermore, source localization results differ between text and table. All together, it remains unclear what the results contribute to the understanding and future conduction of alpha neurofeedback training and its effectiveness to enhance cognitive performance. Therefore, the manuscript needs to be substantially revised before to be considered for publication.

Major points:

1. It is not clear on what basis the research question was derived and why it is important to explore it. The introduction is rather confusing and it is not convincingly reported why the authors investigated dipole sources in EEG activity that was measured after neurofeedback training and not during training. Transfer of neurofeedback training to resting-state activity in previous studies has been mixed and EEG activity after training could simply display fatigue and not the training effect itself (as there was no feedback during the alpha blocks of the whole-head EEG). Moreover, it is not apparent why and how the authors chose the four analysis techniques and compared them. I was wondering why the authors did not use one of the more conventional source localization methods like beamforming or LORETA. The abstract even mentions that alpha neurofeedback training is important for memory function but this focus (or generally transfer of neurofeedback training) is not explored at all (only mentioned in the discussion, please see point 6).

2. Many questions remain throughout the methods section that make me wonder whether the used methods were the most appropriate ones to answer the research question:

- What is the three-dipole model that was mentioned in the abstract but never explained in the manuscript?

- Was the neurofeedback training on fixed days or were the three sessions per week conducted randomly?

- Why was feedback to EEG activity over central sites given during neurofeedback training? On what basis were these electrodes chosen? What does the chosen neurofeedback protocol mean for other alpha neurofeedback studies that give feedback to other electrode sites?

- Why was resting-state EEG only investigated for responders? How many participants were responders?

- Did the study have a different aim during conduction – why would you need the control group then?

- Did participants use strategies during resting-state (p. 5, lines 103-105) and if so, why?

- Was the frequency band for the control group different for every session or did it repeat by chance?

- What were the criteria for artifact removal (p. 6, line 163)? How were eye and muscle artifacts controlled for during neurofeedback training? How much data had to be removed for analysis?

- What did participants do during the 5-minute breaks of the whole-head EEG measurement?

- What does a positive / negative alpha peak (p.7, lines 171-178)? Would an alpha increase/decrease be a more fitting description?

- How many trials of the whole-head EEG measurement were used for each analysis method? Why did the number of trials differ for the four analysis methods (p. 16)?

3. Topographic results are reported only on a visual basis (p. 9)! Please report statistical analyses if you want to discuss differing topographies. Similarly, the source localization results were statistically compared between methods (pp. 10-11) but localized brain regions were not tested statistically against each other within the same method (e.g., is the source localization of the precuneus, PCC, and middle temporal gyrus detected by the positive average significantly greater than zero or greater than detection of other regions using the same method?).

4. The source localization results reported in the results section (p. 10) differ from the results displayed in Table 1 (p. 11). The authors state that 32%-84% of the participants showed dipoles in the reported regions of the precuneus, PCC, and middle temporal gyrus. However, the results in the text are bilateral and the results in the table are split for left and right brain regions. How can you exclude that a participant showing a dipole in right precuneus didn’t also show a dipole in left precuneus? The authors simply added the number of participants for left and right dipoles together. Looking at the results split for left and right brain regions, in some regions barely showed a dipole (4%-16%, which corresponds to 1-4 participants).

5. Considering that 63 participants took part in the study in total, the source localization results rely on a very small number of participants (1-14 participants for the localized brain regions). What does that mean for the generalization of the results? I doubt that effect sizes (that were not reported) are considerably large.

6. The authors tried to integrate their results in the current literature in the discussion. However, it is not clear how the source localization in the present study specifically contributes to the understanding of alpha neurofeedback training. Specifically, the conclusion about a “global alpha activity” (p. 17) that spreads from parietal to frontal regions is highly speculative and is not supported by the results (sources in parietal regions solely). Furthermore, the interpretation that sources of alpha activity after neurofeedback training are regions important for memory because of the strategies that participants used lacks any evidential basis. If a conclusion about memory performance based on neurofeedback is drawn, the authors should systematically investigate transfer to memory tasks or explore the nature of used strategies during training and the following whole-head EEG measurement.

Reviewer #3: Dear authors,

I am sorry to inform you that I did not reccomend your study for publication.

I realise that you performed an extensive study with 12 sessions of neurofeedback training and your neurofeedback data looks very promising (although you should use an identical instruction for the experimental and the control group, to make sure the alpha increase originates from NFT and not from your instruction to think of positive memories (which in itself leads to relaxation and to increased alpha)).

I also really liked your camera installation during neurofeedback training, that is a great method!

The main concern I have with your study is the attempt to perform EEG source localisation with an 32 electrode EEG cap. Here is a paper you may want to look at (https://www.sciencedirect.com/science/article/pii/S0165027015003064), where EEG data was simulated and the validity of source localisation with different electrode densities was tested. Everything below 64 electrodes was simply inaccurate, but even 64 electrodes wasn't good. It should not be performed with anything below 128 electrodes (and even then usually T1 scans of each participant are used).

If you want to use your dataset for future publications, please also make sure you do present correct units in every figure and also do not mix up amplitude and power in the manuscript.

Best wishes and all the best

Marion Brickwedde

Reviewer #4: The present study aimed to identify sources of training-induced alpha activity through four different temporal/spectral analytic techniques, relating the effects of the training to ”memory function”.

As I understood, the authors have already reviewed their paper. In my opinion, the authors did a very good work, I also guess thanks to the valuable reviewers' feedback.

The paper now is very well structured, with a properly deep introduction and large and detailed discussions.

Undoubtedly, I believe some weaknesses still affect the work, most of all the generic reference of alpha power in their study to memory functions without directly testing this reference by means of a specific memory task used in memory studies carried out in the context of cognitive neuroscience literature on this matter.

Indeed, I was most surprised of finding this reference in the light of the kind of procedure to whom their experimental participants were submitted, namely a neurofeedback training (NFT) to self-regulate their own brain activity. Indeed, rather than a mnemonic task, this can be referred to as a learning to self-regulate their own brain alpha activity by means of a progressive increase of individual cognitive control on brain neural networks subserving alertness and focusing of attention functions.

However, I believe that this may not be hard to solve now. Indeed, I think that the hard work made so far by the authors would deserve one more chance to achieve an acceptable level and be published.

Indeed, the EEG literature is plenty of studies directly relating alpha power to the functions of alerting, orienting, and cognitive control characterizing the different neural networks of the visual selective attention system. Below, for instance, I reported a short list of recent and less recent studies in the literature on alpha power and visual attention to be used as a starting point for a critical review of their findings and of those papers referred to by these studies:

Zani, A.; Tumminelli, C.; Proverbio, A.M. Electroencephalogram (EEG) Alpha Power as a Marker of Visuospatial Attention Orienting and Suppression in Normoxia and Hypoxia. An Exploratory Study. Brain Sci. 2020, 10, 140.

Misselhorn, J.; Friese, U.; Engel, A.K. Frontal and parietal alpha oscillations reflect attentional modulation of cross-modal matching. Sci. Rep. 2019, 22, 5030.

Rihs, T.A.; Michel, C.M.; Thut, G. Mechanisms of selective inhibition in visual spatial attention are indexed by alpha-band EEG synchronization. Eur. J. Neurosci. 2017, 25, 603–610.

Poch, C.; Carretie, L.; Campo, P. A dual mechanism underlying alpha lateralization in attentional orienting to mental representation. Biol. Psychol. 2017, 28, 63–70.

Foxe, J.J.; Snyder, A.C. The role of alpha-band brain oscillations as a sensory suppression mechanism during selective attention. Frnt. Hum. Neurosci. 2011, 2, 154.

Kelly, S.P.; Lalor, E.C.; Reilly, R.B.; Foxe, J.J. Increases in Alpha Oscillatory Power Reflect a Suppression during Sustained Visuospatial Attention Active Retinotopic Mechanism for Distracter. J. Neurophysiol. 2006, 95, 3844–3851.

As I already wrote, I believe that if the authors will be able to eliminate any reference to generic and abstract “memory functions” both in the Abstract and in any other paper sections, mostly the Introduction and Discussion sections, and to provide a short critical discussion of their NFT alpha findings in the light of the attentional findings advanced in the indicated studies, the paper will be more than worth to be published.

As a more specific point, please provide a very short explanation of the NFT procedure already in the Abstract.

6. PLOS authors have the option to publish the peer review history of their article (what does this mean?). If published, this will include your full peer review and any attached files.

Reviewer #1: No

Reviewer #2: No

Reviewer #3: **Yes: **Marion Brickwedde, PhD

Reviewer #4: No

---

## [Author Response · Author response to Decision Letter 0]

28 Jan 2021

PONE-D-20-20804

Equivalent current dipole sources of neurofeedback training-induced alpha activity through temporal/spectral analytic techniques.

PLOS ONE

Dear Dr. Shaw,

Thank you for submitting your manuscript to PLOS ONE. After careful consideration, we feel that it has merit but does not fully meet PLOS ONE’s publication criteria as it currently stands. Therefore, we invite you to submit a revised version of the manuscript that addresses the points raised during the review process.

We look forward to receiving your revised manuscript.

Kind regards,

Alice Mado Proverbio

Academic Editor

PLOS ONE

Journal Requirements:

2. In your Methods section, please provide additional information about the participant recruitment method and the demographic details of your participants. Please ensure you have provided sufficient details to replicate the analyses such as: a) the recruitment date range (month and year), b) a description of any inclusion/exclusion criteria that were applied to participant recruitment, c) a table of relevant demographic details, d) a statement as to whether your sample can be considered representative of a larger population, e) a description of how participants were recruited, and f) descriptions of where participants were recruited and where the research took place.

Response: Thank you for the editor’s suggestion. In this version, we have provided demographic details (Table 1) and other information mentioned above in the “Materials and methods” section.

(lines 89-95) “In this study, the participants recruited via social media, e.g., Facebook and Instagram, included 63 healthy students studying at National Cheng Kung University from May 2016 to March 2018 using convenience sampling, and they were randomly assigned to two age- and sex-matched groups. The inclusion criteria were Taiwanese nationality, 20-30 years old, right-handed and not having participated in an NFT study in the past. The exclusion criteria included a history of mental or neurological disorders and potential pregnancy.” 

(line 122) “All of the experiments were performed in our laboratory.”

3. Thank you for your ethics statement:

'The experimental procedure was reviewed and approved by a local research ethics committee. Informed consent was provided and signed by all participants before the experiment.'

Response: Thank you for the editor’s suggestion. In this version, the full name of the ethics committee/institutional review board is given as the Institutional Review Board of the National Cheng Kung University Hospital (NCKUH-IRB). (lines 101-103)

"The authors thank the following institutions for their research support: Ministry of Science and Technology, Taiwan (108-2410-H-006-112-MY3 and 108-2634-F-006-012), and the Mind Research and Imaging Center, National Cheng Kung University, Taiwan."

Response: Thank you for the editor’s suggestion. In this version, we have removed funding information from the “Acknowledgments” section.

Reviewers' comments:

Reviewer's Responses to Questions

Comments to the Author

1. Is the manuscript technically sound, and do the data support the conclusions?

Reviewer #1: Partly

Reviewer #2: Partly

Reviewer #3: No

Reviewer #4: Partly

2. Has the statistical analysis been performed appropriately and rigorously? 

Reviewer #1: Yes

Reviewer #2: No

Reviewer #3: I Don't Know

Reviewer #4: Yes

3. Have the authors made all data underlying the findings in their manuscript fully available?

Reviewer #1: No

Reviewer #2: Yes

Reviewer #3: Yes

Reviewer #4: Yes

4. Is the manuscript presented in an intelligible fashion and written in standard English?

Reviewer #1: Yes

Reviewer #2: Yes

Reviewer #3: Yes

Reviewer #4: Yes

5. Review Comments to the Author

Reviewer #1: The manuscript titled “Equivalent current dipole sources of neurofeedback training-induced alpha activity through temporal/spectral analytic techniques” touches upon evergreen subject of EEG neurofeedback. It comes with reasonable number of participants (63) and an interesting attempt to localize sources of observed EEG activity.

Although interesting some major questions need to be answered before final conclusions can be made:

1. The training protocol was based on the alpha band which is highly susceptible to various manipulation (Williams, 1977; Quandt et al. 2012) and the training protocol included instructions for participants. How many trainers did provide those instructions and how authors assured mitigation of trainer effect?

Response: Thank you for the reviewer’s suggestion. In this study, only one trainer provided instructions and strategies and trained all of the participants. Before the experiment, the trainer was trained to be familiar with the training protocol, ensuring that the trainer could provide instructions and strategies to each participant consistently during training.

2. Were the participants personality profiles investigated (using psychological tests such as EPQ-R, NEO-EFI,…) ?

Response: Thank you for the reviewer’s suggestion. In this study, we did not investigate the participants’ personality profiles. This issue is an interesting one for future NFT research.

Authors’ results are based only on the “responders” subgroup of the “alpha” - experimental group：

1. How many exactly “responders” were finally identified?

Response: Thank you for the reviewer’s suggestion. In this study, twenty-nine participants (83%) were identified as “responders.” (line 261).

2. Are there any specific features allowing to distinguish responders vs non-non-responders other than increase in alpha amplitude?

Response: Thank you for the reviewer’s suggestion. Yes, in our previous study [1], we used total alpha duration (all successful 1-s events within a session were accumulated) to distinguish responders.

3. Is there any relations between responders, non-responders and instructions, trainers or personality traits?

Response: Thank you for the reviewer’s suggestion. In this study, only one trainer trained all of the participants with consistent instructions. We did not investigate the participants’ personality profiles. There were no relations of responders and non-responders with instructions, trainers or personality traits. This issue is also another interesting one for future NFT research.

4. What would be result for comparison all “alpha” (experimental) group vs “ctrl” (control) group?

Response: Thank you for the reviewer’s suggestion. No significant difference was observed in mean alpha amplitude of the Ctrl group throughout the 12 sessions. Only the all-“alpha” (experimental) group showed a progressively increasing change in alpha amplitude throughout the training sessions. The result (see below) was similar to that in Fig. 1. We focused on well-trained “responders” in the present study. Therefore, we only presented the data of “responders” in the Alpha group.

Mean relative alpha amplitude changes throughout the 12 sessions in the all-“Alpha” (experimental) group and “Ctrl” (control) group. The gray line is a linear regression line and indicates a progressive, linear increase in the Alpha group. * p < 0.05 vs. the 1st session. # p < 0.05 vs. the Ctrl group.

Authors used bipolar electrode montage very rarely used in this kind of investigations which makes it difficult to compare to similar studies:

1. Why Authors decided to use bipolar montage?

Response: Thank you for the reviewer’s suggestion. In the present study, a bipolar electrode montage was only used in the neurofeedback training and not whole-head EEG. Bipolar recording was beneficial to reduce the possible artifacts of motion or eye blinks. The purpose (either bipolar montage or other montage of other NFT studies) is to calculate changes in alpha activity during training. It would be interesting to investigate differences between montages for future research.

2. What would be the results recalculated to the average reference?

Response: Thank you for the reviewer’s suggestion. The EEGs of six electrodes were converted into three bipolar EEGs in the amplifier circuit board. Three bipolar EEGs were digitized by an analog-to-digital converter to computer and were acquired by LabVIEW software. Therefore, there are no original EEGs of six electrodes that can be recalculated to the average reference.

3. What was the results of whole head analyses for “non-responders”?

Response: In this study, only well-trained “responders” in the Alpha group participated in whole-head EEG recording. “Non-responders” did not meet the criteria of “responders” and did not participant in whole-head EEG recordings. Therefore, there are no results of whole head analyses for “non-responders.”

Source analyses

1. Sources were calculated using Curry software which the procedure is not publicly available, could you provide detailed methodology, please?

Response: There have been many studies using Curry software (transition from Scan software) to conduct dipole source localization. Generally, of course, a qualified signal is needed first; the procedure includes deciding on the number of dipoles used for dipole source localization (depending on assumptions according to neurophysiological and anatomical knowledge from previous studies) and choosing a head model (e.g., spherical or BEM) and dipole model (e.g., regional or LORETA). However, no recent study has provided information about the sources of training-induced alpha activity. Therefore, we used ICA to define the numbers of dipoles. The number of dipoles used for dipole source localization depended on the number of valid components (SNR > 1) [20]. Dipole source localization was conducted for each “responder.” Subsequently, we projected the dipoles onto the brain.

2. From my understanding sources were calculated using ICA. Since ICA will return different clusters for each participant how the results were averaged: calculated individually, finding common clusters and then averaged or they were all subjects were analyzed together?

Response: Thank you for the reviewer’s suggestion. The ICA and dipole source localization was performed for each “responder.” In this study, the purpose of using ICA was simply to define the numbers of dipoles used in dipole source localization. For each “responder,” the averaged alpha epoch was applied in ICA before dipole source localization. Several ICA components (depending on the number of channels) were estimated, and we calculated their SNR. The number of dipoles used for dipole source localization depended on the number of valid components (SNR > 1). When the number of dipoles was decided, it was used for dipole source localization of the averaged alpha epoch. In this revised version, we have reorganized the description in the dipole source localization part of the “Materials and methods” section.

(lines 215-222) “In the present study, independent component analysis (ICA) was used to define the numbers of dipoles, and dipole source localization was performed for each “responder.” ICA source reconstruction was conducted to compute a fixed dipole source for each independent component [19]. The averaged alpha epoch was applied in ICA before dipole source localization. ICA components were estimated, and we calculated their signal-to-noise ratio (SNR). The number of dipoles used for dipole source localization depended on the number of valid components (SNR > 1) [20].”

3. Without individual electrode positions and/or individual T1s localization results will not provide accurate results allowing for anatomical considerations

Response: Thank you for the reviewer’s suggestion. We had individual electrode positions. In this study, all of the electrode positions were measured using a 3D digitizer (Fastrak, Polhemus, USA) for dipole source analyses. The description has been added to the whole-head EEG and processing part of the “Materials and methods” section (lines 169-170). In contrast, we did not acquire individual 3D MRI data but used standardized BEM models to conduct source localization. Indeed, individual MRI data can improve the source localization accuracy. However, source localization using EEG is used to estimate approximate brain regions. Fuchs et al. (2004) (https://doi.org/10.1016/S1388-2457(02)00030-5) reported that no significant differences in source localization were observed and showed the similarity in source localization between individual and standardized BEM models. Moreover, the standardized BEM models showed significantly better localization accuracy than the individual BEM model. The evidence revealed that standardized BEM models remain sufficient for source localization. Certainly, there are several factors affecting the accuracy of source localization, e.g., different brain activities, head models and source localization methods. For training-induced alpha activity, these issues would be interesting for future research.

Other methodological questions:

1. Authors investigated also the delta, theta and beta (13-30 Hz) bands what was the results how they compare to alpha band results (especially the theta band which could possibly drive the alpha one)?

Response: Regarding the reviewer’s concern, we calculated the amplitude difference between 12th and 1st sessions for each band. The paired t-test revealed that the amplitude difference of the alpha band was significantly greater than that of delta (p = 0.005), theta (p = 0.007) and beta (p < 0.001). No significant amplitude differences were observed between delta and theta (p = 0.176), delta and beta (p = 0.573), and theta and beta (p = 0.291). The results were the same as the results when we compared the amplitudes of these bands between the 1st and 12th sessions (lines 282-284). These data indicated that no amplitude of bands was increased, except for the alpha band after NFT.

2. What was exactly calculation of ERSPA?

Response: The calculation of ERSPA used the ERSP technology to measure amplitude spectra at each time point for each categorized alpha epoch, and then amplitude spectra were averaged over time. An inverse FFT was applied to averaged ERSP to generate a time series of averaged alpha epoch. The description has been added to the whole-head EEG and processing part of the “Materials and methods” section.

(lines 199-202) “The ERSP technology measured amplitude spectra at each time point. The amplitude spectra of each categorized alpha epoch were averaged over time. An inverse FFT was applied to the averaged ERSP to generate a time series of averaged alpha epoch.”

Reviewer #2: The authors aimed to explore the neurophysiological sources of alpha increases induced by neurofeedback training. Therefore, they calculated the power increases during 12 sessions of alpha neurofeedback training by using four different analysis techniques, namely the maximum peak average, the positive average, the negative average, and the event-related spectral perturbation average. All analysis methods showed a high agreement on localized dipole sources in the precuneus, posterior cingulate cortex, and middle temporal gyrus. The authors conclude that all measures are suitable for investigating dipole sources during neurofeedback training.

The research question itself is an interesting one and worth to be investigated. However, it is unclear how the authors derived the research question and why the described methods were chosen in order to answer it. The method section lacks important descriptions and it is not comprehensible why and on the basis of which criteria the authors ended up with a source localization analysis with 14 participants despite 63 that were tested. It is questionable why four analysis techniques were compared, considering that the research question focused on exploration of the source regions of alpha activity, and it is not reported whether source localization would have been significant using solely one of these methods. Furthermore, source localization results differ between text and table. All together, it remains unclear what the results contribute to the understanding and future conduction of alpha neurofeedback training and its effectiveness to enhance cognitive performance. Therefore, the manuscript needs to be substantially revised before to be considered for publication.

Major points:

1. It is not clear on what basis the research question was derived and why it is important to explore it. The introduction is rather confusing and it is not convincingly reported why the authors investigated dipole sources in EEG activity that was measured after neurofeedback training and not during training. Transfer of neurofeedback training to resting-state activity in previous studies has been mixed and EEG activity after training could simply display fatigue and not the training effect itself (as there was no feedback during the alpha blocks of the whole-head EEG). Moreover, it is not apparent why and how the authors chose the four analysis techniques and compared them. I was wondering why the authors did not use one of the more conventional source localization methods like beamforming or LORETA. The abstract even mentions that alpha neurofeedback training is important for memory function but this focus (or generally transfer of neurofeedback training) is not explored at all (only mentioned in the discussion, please see point 6).

Response: 

Regarding the reviewer’s concerns, we have reorganized the introduction to clarify the research question (lines 36-79). The purpose of this study was to investigate the generation of training-induced alpha activity. During training, participants still learned how to successfully enhance their alpha activity. As in Fig. 1, alpha amplitude still did not significantly increase in the initial training session, showing that the participants were still unstable to enhance alpha activity. After 12 training sessions, we found that most of the participants could successfully enhance and increase alpha activity. Furthermore, we chose “responders” to conduct whole-head EEG recording and extract successful alpha trials (> 1.5 averaged alpha amplitude of EEGs of fixation blocks) for dipole source localization. Indeed, there was no feedback during whole-head EEG recording. However, our “responders” reported that they had learned a strategy to enhance alpha activity after NFT. They had enhanced alpha activity in alpha blocks of whole-head EEG, resulting in approximately 250-500 successful alpha trials (each “responder”), which were used for analysis. Indeed, the temporal impact of NFT on brain activity is an interesting issue for future NFT research.

There are two types of localization methods. One is a distributed method (such as LORETA or beamformer), and the other one is a dipolar method (such as regional dipole). The disadvantages of LORETA are low spatial resolution and blurred localized images of a point source with dispersion in the image (Asadzadeh et al., 2020) (https://doi.org/10.1016/j.jneumeth.2020.108740). The distributed method is an area-wise localization. In contrast, the dipolar method is a point-wise localization. When the dipole number is unknown, the dipolar method is suitable for source localization (Sonoda et al., 2018) (https://doi.org/10.1016/j.neunet.2018.08.008). There are still no assumptions helping us to decide the dipole number used for dipole fitting in training-induced alpha activity. Therefore, we used the regional dipole model in the present study. It would be interesting investigate dipole localization of training-induced alpha activity by different methods in future research.

Thanks for the reviewer’s constructive suggestion. You mentioned that we did not explore the memory function in this study. We agree and have corrected the description in the abstract (lines 2-3). In addition, we have also reorganized the discussion in the “Discussion” section (please see point 6).

(lines 2-3) “Much of the work in alpha NFT has focused on evaluating changes in alpha amplitude.”

2. Many questions remain throughout the methods section that make me wonder whether the used methods were the most appropriate ones to answer the research question:

- What is the three-dipole model that was mentioned in the abstract but never explained in the manuscript?

Response: Thank you for the reviewer’s suggestion. No recent studies have provided information about the sources of training-induced alpha activity. In this study, we found that sources of training-induced alpha activity were mostly clustered in the precuneus, PCC and middle temporal gyrus. These three regions would be good candidates for dipole source localization, which could help researchers to decide on the number of dipoles in future studies. We have corrected “three-dipole model” to “three dipoles” in the abstract (lines 24).

- Was the neurofeedback training on fixed days or were the three sessions per week conducted randomly?

Response: Thank you for the reviewer’s suggestion. The neurofeedback training was performed on fixed days and at fixed times. The days and times were chosen by discussing them with the participants.

- Why was feedback to EEG activity over central sites given during neurofeedback training? On what basis were these electrodes chosen? What does the chosen neurofeedback protocol mean for other alpha neurofeedback studies that give feedback to other electrode sites?

Response: Previous alpha NFT studies have reported that alpha activity can be modulated using C3, Cz and C4 electrodes (Lecomte et al., 2011, https://doi.org/10.4236/psych.2011.28129; Nan et al., 2012, https://doi.org/10.1016/j.ijpsycho.2012.07.182.; Wei et al., 2017, https://doi.org/10.1186/s12938-017-0418-8.). In our previous study [1], we found that alpha activity was enhanced after NFT when electrodes located centrally (2.5 cm anterior and posterior to C3, Cz, and C4). The evidence indicated that these electrodes were good candidates for alpha NFT. Therefore, in this study, we used the same electrodes as in our previous study. Indeed, it would be interesting to investigate the effects of different electrodes on modulating alpha activity in future research.

- Why was resting-state EEG only investigated for responders? How many participants were responders?

Response: In this study, “responders” were defined as well-trained participants, meaning that they had learned a strategy to enhance their alpha activity successfully. Therefore, twenty-nine participants (83%) in the Alpha group were identified as well-trained “responders” and participated in whole-head EEG recording. The whole-head EEG recording contained fixation (same as the baseline block of neurofeedback training, during which participants were asked to keep their eyes open and not to engage in any training event from the training block) and alpha blocks (producing training-induced alpha activity). We used data of whole-head EEG to characterize the topographic map and dipole source localization of training-induced alpha activity.

- Did the study have a different aim during conduction – why would you need the control group then?

Response: We added the control group, which was compared to the alpha group to test the effect of alpha NFT on alpha enhancement. Both groups showed no significant differences in alpha amplitude at the beginning, indicating that the characteristics of the alpha group were the same as those of the control group. After NFT, the control group showed no change in alpha amplitude. In contrast, the alpha group showed significant differences in alpha amplitude compared to the control group. The results confirmed that our training protocol had effective results for alpha enhancement.

- Did participants use strategies during resting-state (p. 5, lines 103-105) and if so, why?

Response: No. In the resting period, we helped the participants recall the type of strategy that they used to achieve high alpha amplitude from the previous training block. Therefore, in this period, no EEG recording was conducted, and we only talked to the participants. Subsequently, the next training block was conducted.

- Was the frequency band for the control group different for every session or did it repeat by chance?

Response: No, there were no significant differences between sessions. ANOVA revealed no significant main effect of the session on delta (p = 0.179), theta (p = 0.108), alpha (p = 0.157), and beta (p = 0.104).

- What were the criteria for artifact removal (p. 6, line 163)? How were eye and muscle artifacts controlled for during neurofeedback training? How much data had to be removed for analysis?

Response: Thank you for the reviewer’s suggestion.

1. Successful alpha events contaminated by blinks, eye-movement artifacts (> 65 μV) or remarkable muscle activity were automatically removed (lines 184-185).

2. Before neurofeedback training, we asked the participants not to use inadequate strategies involving eye closure or body movements during training. In addition, we used a digital camera to exclude the effects of possible behavioral artifacts (lines 116-119). Bipolar recording was also beneficial to reduce the possible artifacts of motion or eye blinks. Therefore, rare data (< 1%) were removed for analysis.

- What did participants do during the 5-minute breaks of the whole-head EEG measurement?

Response: The participants took a break for 5 minutes without EEG measurements and waited for the next run.

- What does a positive / negative alpha peak (p.7, lines 171-178)? Would an alpha increase/decrease be a more fitting description?

Response: Thank you for the reviewer’s suggestion. We have corrected the description in the whole-head EEG and processing part of the “Materials and methods” section.

(lines 187-190) “The time points of the highest voltage positive peak of the successful alpha events were identified and defined as trigger points. Based on the time points, 1.2-s alpha epochs centered on the positive peak of the artifact-free data were extracted and averaged.”

(lines 195-197) “If the first peak voltage in an alpha epoch was positive or negative, it was categorized as a positive alpha epoch or a negative alpha epoch, respectively.”

- How many trials of the whole-head EEG measurement were used for each analysis method? Why did the number of trials differ for the four analysis methods (p. 16)?

Response: When a 1-s EEG alpha amplitude of the alpha blocks was higher than the 1.5-fold averaged alpha amplitude of all of the 1-s fixation EEGs, the 1-s EEG segment (epoch) was considered a successful alpha event (trials). For each “responder,” approximately 250-500 trials were used for analysis. All of the trials were used for the MPA and ERSPA. However, to identify whether the first peak in a trial was positive or negative, all of the trials were assigned to positive trials and negative trials. Therefore, the numbers of trials for PA and NA methods were smaller than for MPA and ERSPA methods.

3. Topographic results are reported only on a visual basis (p. 9)! Please report statistical analyses if you want to discuss differing topographies. Similarly, the source localization results were statistically compared between methods (pp. 10-11) but localized brain regions were not tested statistically against each other within the same method (e.g., is the source localization of the precuneus, PCC, and middle temporal gyrus detected by the positive average significantly greater than zero or greater than detection of other regions using the same method?).

Response: Thank you for the reviewer’s constructive suggestion. In this version, we have evaluated differences among the alpha amplitudes of frontal (FP1, FP2, F3, FZ, F4 FC3, FCZ and FC4), parietal (CP3, CPZ, CP4, P3, PZ and P4), occipital (O3, OZ and O4), frontotemporal (F7, F8, FT7 and FT8), and parietotemporal (TP7, TP8, T5 and T6) regions with each method (lines 207-211). The statistical results have been added in the EEG mapping of training-induced alpha activity part of the “Results” section (lines 294-323). The percentage of “responders” in each localized brain region are tested statistically against zero in each method. The results have been added to the dipole source localization part of the “Results” section (lines 342-355).

(lines 207-211) “Furthermore, to evaluate the alpha-dominant region, the alpha amplitudes of the averaged alpha epoch of the electrodes close to the frontal (FP1, FP2, F3, FZ, F4 FC3, FCZ and FC4), parietal (CP3, CPZ, CP4, P3, PZ and P4), occipital (O3, OZ and O4), frontotemporal (F7, F8, FT7 and FT8), and parietotemporal (TP7, TP8, T5 and T6) regions were averaged separately.”

(lines 294-323) “For the MPA method, the highest peak in the middle of the alpha epoch and a flat period on both sides were observed. Significant differences among the alpha amplitudes of the parietal and frontal (t = 2.137, p = 0.041), parietal and occipital (t = 6.558, p < 0.001), parietal and frontotemporal (t = 6.583, p < 0.001), parietal and parietotemporal (t = 7.335, p < 0.001), frontal and frontotemporal (t = 15.239, p < 0.001), frontal and parietotemporal (t = 2.315, p = 0.028), occipital and frontotemporal (t = 3.680, p = 0.001), and parietotemporal and frontotemporal (t = 3.640, p = 0.001) regions were observed. For the PA method, the largest positive peak followed the start of the alpha epoch (trigger point). Significant differences among the alpha amplitudes of the parietal and frontal (t = 2.340, p = 0.027), parietal and occipital (t = 5.699, p < 0.001), parietal and frontotemporal (t = 4.521, p < 0.001), parietal and parietotemporal (t = 4.555, p < 0.001), frontal and occipital (t = 1.793, p = 0.007), frontal and frontotemporal (t = 2.778, p = 0.010), occipital and frontotemporal (t = -0.784, p = 0.035), and occipital and parietotemporal (t = -1.055, p = 0.001) regions were observed. For the NA methods, the negative peak followed the start of the alpha epoch (trigger point). Significant differences among the alpha amplitudes of the parietal and frontal (t = 1.356, p = 0.020), parietal and occipital (t = 4.518, p < 0.001), parietal and frontotemporal (t = 4.424, p < 0.001), parietal and parietotemporal (t = 5.217, p < 0.001), frontal and occipital (t = 1.627, p = 0.040), frontal and frontotemporal (t = 5.969, p < 0.001), occipital and parietotemporal (t = -0.824, p = 0.032), and occipital and frontotemporal (t = 1.003, p = 0.042) regions were observed. Before the trigger point, a flat period was observed with the PA and NA methods. For the ERSPA method, alpha activity was obviously observed after inverse FFT implementation. Significant differences among the alpha amplitudes of the parietal and frontal (t = 0.714, p = 0.045), parietal and occipital (t = 1.077, p = 0.003), parietal and frontotemporal (t = 1.751, p = 0.002), parietal and parietotemporal (t = 2.568, p = 0.002), frontal and occipital (t = 0.622, p = 0.042), frontal and frontotemporal (t = 1.332, p = 0.049), occipital and parietotemporal (t = 0.299, p = 0.026), and parietotemporal and frontotemporal (t = 0.195, p = 0.028) regions were observed. These data indicated that training-induced alpha activity in the parietal region was significantly higher than other regions.”

(lines 342-355) “For the MPA method, the percentages of “responders” whose dipoles were clustered in the precuneus, PCC and middle temporal areas were 84% (p < 0.001), 68% (p < 0.001) and 40% (p < 0.001) significantly different from zero, respectively. For the PA method, the percentages of “responders” whose dipoles were clustered in the precuneus, PCC and middle temporal areas were 72% (p < 0.001), 36% (p = 0.001) and 48% (p < 0.001) significantly different from zero, respectively. For the NA method, the percentages of “responders” whose dipoles were clustered in the precuneus, PCC and middle temporal areas were 64% (p < 0.001), 40% (p = 0.005) and 40% (p < 0.001) significantly different from zero, respectively. For the ERSPA method, the percentages of “responders” whose dipoles were clustered in the precuneus, PCC and middle temporal areas were 64% (p < 0.001), 32% (p = 0.002) and 44% (p < 0.001) significantly different from zero, respectively. The percentage of “responder” whose dipoles were located in the other regions was ≤ 16% for each method and showed no significant different from zero.”

4. The source localization results reported in the results section (p. 10) differ from the results displayed in Table 1 (p. 11). The authors state that 32%-84% of the participants showed dipoles in the reported regions of the precuneus, PCC, and middle temporal gyrus. However, the results in the text are bilateral and the results in the table are split for left and right brain regions. How can you exclude that a participant showing a dipole in right precuneus didn’t also show a dipole in left precuneus? The authors simply added the number of participants for left and right dipoles together. Looking at the results split for left and right brain regions, in some regions barely showed a dipole (4%-16%, which corresponds to 1-4 participants).

Response: Thank you for the reviewer’s constructive suggestion. In this study, dipoles of each “responder” were located in different regions. The dipole locations were also different between “responders.” In this revised version, we have removed the term “bilateral” from the text. The number (N) of “responders” in Table 2 indicates the number of “responders” whose dipoles are located in the brain regions, whether in one or both hemispheres. For example, if one dipole is located in the right precuneus or one in the right and one in the left, the number (N) increased by 1.

5. Considering that 63 participants took part in the study in total, the source localization results rely on a very small number of participants (1-14 participants for the localized brain regions). What does that mean for the generalization of the results? I doubt that effect sizes (that were not reported) are considerably large.

Response: Thank you for the reviewer’s suggestion. In this study, only well-trained “responders” in the Alpha group participated in whole-head EEG recording (lines 260-262). The whole-head EEG data of twenty-five “responder” were used to perform dipole source localization. The number (N) of “responders” in Table 2 indicates how many “responders” whose dipoles were located in the brain regions, whether in one or both hemispheres. Therefore, the percentage of “responders” whose dipoles were clustered in the precuneus (64%-84%), PCC (36%-68%) and middle temporal areas (40%-48%) were significantly different from zero among the four methods (lines 342-355). The results indicated that the precuneus, PCC and middle temporal gyrus seem to play important roles in enhancing training-induced alpha activity.

6. The authors tried to integrate their results in the current literature in the discussion. However, it is not clear how the source localization in the present study specifically contributes to the understanding of alpha neurofeedback training. Specifically, the conclusion about a “global alpha activity” (p. 17) that spreads from parietal to frontal regions is highly speculative and is not supported by the results (sources in parietal regions solely). Furthermore, the interpretation that sources of alpha activity after neurofeedback training are regions important for memory because of the strategies that participants used lacks any evidential basis. If a conclusion about memory performance based on neurofeedback is drawn, the authors should systematically investigate transfer to memory tasks or explore the nature of used strategies during training and the following whole-head EEG measurement.

Response: Response: We agree and thank you for the reviewer’s constructive suggestion. Previous alpha NFT studies reported that cognitive performance was improved after alpha NFT. These findings indicated that participants learned to self-regulate their brains to increase alpha activity, which could improve their cognitive performance. However, there are no studies investigating which regions of the brain work when enhancing alpha activity. If we can determine the sources of the training-induced alpha activity, it could provide valuable information to support cognitive improvement after alpha NFT. Indeed, we did not investigate the effect of alpha NFT on memory performance. In the revised version, we have reorganized and added an extensive discussion to the “Discussion” section (lines 468-489). Furthermore, we also have evaluated differences among the alpha amplitudes of the frontal, parietal, occipital, frontotemporal, and parietotemporal electrodes with each method. The results indicated that the training-induced alpha activity of the parietal region was significantly higher than in the other regions. We have reorganized this discussion in the “Discussion” section (lines 490-507). 

 (lines 468-489) “Alpha activity has been thought to be associated with attention function. For example, alpha power increases were found during internal attention tasks [30] and decreases during demanding arithmetic tasks [31]. In the present study, participants learned to self-regulate alpha activity by shifting internal (mental imagery) and external attention (feedback) during training. Such a shift produces comparable signatures of alpha activity modulations [32]. Alpha activity increases reflected internal attention of mental imagery [33]. In neuroimaging studies, the precuneus is involved in visuospatial attention functions [34]. The PCC regulates the focus of attention and controls the balance between internal and external attention [35]. Lesions of the middle temporal gyrus were associated with spatial attention deficits in spatial neglect patients [36]. All of the evidence supports the existence of a range of attention-related influences on activation in the precuneus, PCC and middle temporal gyrus. The amplitude of alpha increases reflects cortical inhibition [37]. Inhibition of these three brain regions could protect reorienting against irrelevant stimulation, which could modulate attention function more sufficiently. Consequently, after training, participants might efficiently modulate internal attention to increase alpha activity. Internal attention involves modulation, selection, and maintenance of internally produced information, such as long-term memory or working memory [38]. Considered together, alpha NFT could potentially modulate these three attention-related regions to improve internal attention function and further affect memory performance. Our findings could provide valuable information to partially support memory improvement after alpha NFT in previous studies.”

(lines 490-507) “In the present study, a larger alpha amplitude was observed in the parietal region than in other regions, indicating that training-induced alpha activity dominated in the parietal region. The result was similar with the dipole locations (the PCC and precuneus). In addition, alpha activity also appeared slightly higher in the frontal region. Alpha NFT was thought to modulate top-down control networks, resulting in participants learning successfully to self-regulate their alpha activity. An increased frontal alpha amplitude might enable tight functional coupling between frontal cortical areas, thereby allowing for control of visual processing [39]. Alpha oscillations of frontal brain areas seem to be important for top-down control [4041]. A recent simultaneous EEG-fMRI study further found that long-range alpha synchrony was intrinsically linked to activity in the frontal-parietal control network [42]. This evidence could support frontal alpha activity being slightly enhanced by alpha NFT. Another possible reason for increased frontal alpha amplitudes might be “neural efficiency.” NFT could cause the neurons of the frontal region to integrate information over space and time repeatedly, causing “neural efficiency” [43]. Neural efficiency hypothesizes that neural activation is decreased in experts. Well-trained “responders” as experts had learned to decrease neural activation of frontal region after NFT, resulting in increased frontal alpha activity.”

Reviewer #3: Dear authors,

I am sorry to inform you that I did not recommend your study for publication.

I realize that you performed an extensive study with 12 sessions of neurofeedback training and your neurofeedback data looks very promising (although you should use an identical instruction for the experimental and the control group, to make sure the alpha increase originates from NFT and not from your instruction to think of positive memories (which in itself leads to relaxation and to increased alpha)).

I also really liked your camera installation during neurofeedback training, that is a great method!

The main concern I have with your study is the attempt to perform EEG source localization with an 32 electrode EEG cap. Here is a paper you may want to look at (https://www.sciencedirect.com/science/article/pii/S0165027015003064), where EEG data was simulated and the validity of source localization with different electrode densities was tested. Everything below 64 electrodes was simply inaccurate, but even 64 electrodes wasn't good. It should not be performed with anything below 128 electrodes (and even then usually T1 scans of each participant are used).

Response: Thank you for the reviewer’s suggestion. Indeed, source localization accuracy is improved when electrode numbers increase. Source localization with EEG is used to estimate approximate brain regions. Rodin and Rodin (1995) reported a satisfactory result when sources of averaged occipital alpha epochs were located over the occipital region using 19 electrodes. Sohrabpour et al. (2015) (https://doi.org/10.1016/j.clinph.2014.05.038) reported that dipoles of each single epileptic spike clustered in similar brain regions when electrode numbers decreased from 128 to 96, 64 and 32. Based on the concept of these findings, we averaged alpha epochs to increase the SNR for each participant but did not the average alpha epochs of all “responders”. After dipole source localization, we found dipoles approximately clustered in the bilateral precuneus, PCC and middle temporal regions. Certainly, different brain activity, head models, source localization methods, electrode numbers and SNRs will affect the accuracy of source localization. For training-induced alpha activity, these issues will be interesting for future research.

If you want to use your dataset for future publications, please also make sure you do present correct units in every figure and also do not mix up amplitude and power in the manuscript.

Response: Thank for reviewer’s suggestion. In the present study, we did not transform the amplitude of FFT, for example, square the amplitude as power. Therefore, the units are the same as the original signal unit (microvolt).

Best wishes and all the best

Marion Brickwedde

Reviewer #4: The present study aimed to identify sources of training-induced alpha activity through four different temporal/spectral analytic techniques, relating the effects of the training to ”memory function”.

As I understood, the authors have already reviewed their paper. In my opinion, the authors did a very good work, I also guess thanks to the valuable reviewers' feedback.

The paper now is very well structured, with a properly deep introduction and large and detailed discussions.

Undoubtedly, I believe some weaknesses still affect the work, most of all the generic reference of alpha power in their study to memory functions without directly testing this reference by means of a specific memory task used in memory studies carried out in the context of cognitive neuroscience literature on this matter.

Indeed, I was most surprised of finding this reference in the light of the kind of procedure to whom their experimental participants were submitted, namely a neurofeedback training (NFT) to self-regulate their own brain activity. Indeed, rather than a mnemonic task, this can be referred to as a learning to self-regulate their own brain alpha activity by means of a progressive increase of individual cognitive control on brain neural networks subserving alertness and focusing of attention functions. 

However, I believe that this may not be hard to solve now. Indeed, I think that the hard work made so far by the authors would deserve one more chance to achieve an acceptable level and be published.

Indeed, the EEG literature is plenty of studies directly relating alpha power to the functions of alerting, orienting, and cognitive control characterizing the different neural networks of the visual selective attention system. Below, for instance, I reported a short list of recent and less recent studies in the literature on alpha power and visual attention to be used as a starting point for a critical review of their findings and of those papers referred to by these studies:

Zani, A.; Tumminelli, C.; Proverbio, A.M. Electroencephalogram (EEG) Alpha Power as a Marker of Visuospatial Attention Orienting and Suppression in Normoxia and Hypoxia. An Exploratory Study. Brain Sci. 2020, 10, 140.

Misselhorn, J.; Friese, U.; Engel, A.K. Frontal and parietal alpha oscillations reflect attentional modulation of cross-modal matching. Sci. Rep. 2019, 22, 5030.

Rihs, T.A.; Michel, C.M.; Thut, G. Mechanisms of selective inhibition in visual spatial attention are indexed by alpha-band EEG synchronization. Eur. J. Neurosci. 2017, 25, 603–610.

Poch, C.; Carretie, L.; Campo, P. A dual mechanism underlying alpha lateralization in attentional orienting to mental representation. Biol. Psychol. 2017, 28, 63–70.

Foxe, J.J.; Snyder, A.C. The role of alpha-band brain oscillations as a sensory suppression mechanism during selective attention. Frnt. Hum. Neurosci. 2011, 2, 154.

Kelly, S.P.; Lalor, E.C.; Reilly, R.B.; Foxe, J.J. Increases in Alpha Oscillatory Power Reflect a Suppression during Sustained Visuospatial Attention Active Retinotopic Mechanism for Distracter. J. Neurophysiol. 2006, 95, 3844–3851.

As I already wrote, I believe that if the authors will be able to eliminate any reference to generic and abstract “memory functions” both in the Abstract and in any other paper sections, mostly the Introduction and Discussion sections, and to provide a short critical discussion of their NFT alpha findings in the light of the attentional findings advanced in the indicated studies, the paper will be more than worth to be published.

Response: Thank you for the reviewer’s constructive suggestion. Indeed, participants learned to self-regulate their own brain alpha activity by shifting internal (mental imagery) and external attention (feedback) during training. Progressively, attention function can increase through NFT. Attention and memory are thought to interact closely. From this point of view, it could provide valuable information to support memory improvement after alpha NFT in previous studies. An extensive discussion has been added to the “Discussion” section.

(lines 468-489) “Alpha activity has been thought to be associated with attention function. For example, alpha power increases were found during internal attention tasks [30] and decreases during demanding arithmetic tasks [31]. In the present study, participants learned to self-regulate alpha activity by shifting internal (mental imagery) and external attention (feedback) during training. Such a shift produces comparable signatures of alpha activity modulations [32]. Alpha activity increases reflected internal attention of mental imagery [33]. In neuroimaging studies, the precuneus is involved in visuospatial attention functions [34]. The PCC regulates the focus of attention and controls the balance between internal and external attention [35]. Lesions of the middle temporal gyrus were associated with spatial attention deficits in spatial neglect patients [36]. All of the evidence supports the existence of a range of attention-related influences on activation in the precuneus, PCC and middle temporal gyrus. The amplitude of alpha increases reflects cortical inhibition [37]. Inhibition of these three brain regions could protect reorienting against irrelevant stimulation, which could modulate attention function more sufficiently. Consequently, after training, participants might efficiently modulate internal attention to increase alpha activity. Internal attention involves modulation, selection, and maintenance of internally produced information, such as long-term memory or working memory [38]. Considered together, alpha NFT could potentially modulate these three attention-related regions to improve internal attention function and further affect memory performance. Our findings could provide valuable information to partially support memory improvement after alpha NFT in previous studies.”

As a more specific point, please provide a very short explanation of the NFT procedure already in the Abstract.

Response: Thank you for the reviewer’s suggestion. We have provided an explanation of the NFT procedure in the abstract.

(lines 7-16) “Thirty-five healthy participants were recruited into an alpha group receiving feedback of 8–12-Hz amplitudes, and twenty-eight healthy participants were recruited into a control group receiving feedback of random 4-Hz amplitudes from the range of 7 to 20 Hz. Twelve sessions were performed within 4 weeks (3 sessions per week). The control group had no change in alpha amplitude. In contrast, twenty-nine participants in the alpha group showed significant alpha amplitude increases and were identified as “responders.” A whole-head EEG was recorded for the “responders” after NFT. The epochs of training-induced alpha activity from whole-head EEG were averaged by four different methods for equivalent current dipole source analysis.”

6. PLOS authors have the option to publish the peer review history of their article (what does this mean?). If published, this will include your full peer review and any attached files.

Do you want your identity to be public for this peer review? For information about this choice, including consent withdrawal, please see our Privacy Policy.

Reviewer #1: No

Reviewer #2: No

Reviewer #3: Yes: Marion Brickwedde, PhD

Reviewer #4: No

---

## [Decision Letter · Decision Letter 1]

23 Apr 2021

PONE-D-20-20804R1

Equivalent current dipole sources of neurofeedback training-induced alpha activity through temporal/spectral analytic techniques.

PLOS ONE

Dear Dr. Shaw,

Thank you for submitting your manuscript to PLOS ONE. After careful consideration, we feel that it has merit but does not fully meet PLOS ONE’s publication criteria as it currently stands. Therefore, we invite you to submit a revised version of the manuscript that addresses all the points raised during the review process.

In particoular I require that you provide full analyses including the non-responders.

We look forward to receiving your revised manuscript.

Kind regards,

Francesco Di Russo, Ph.D.

Academic Editor

PLOS ONE

Reviewers' comments:

Reviewer's Responses to Questions

**Comments to the Author**

1. If the authors have adequately addressed your comments raised in a previous round of review and you feel that this manuscript is now acceptable for publication, you may indicate that here to bypass the “Comments to the Author” section, enter your conflict of interest statement in the “Confidential to Editor” section, and submit your "Accept" recommendation.

Reviewer #1: (No Response)

Reviewer #3: (No Response)

2. Is the manuscript technically sound, and do the data support the conclusions?

Reviewer #1: Partly

Reviewer #3: Partly

3. Has the statistical analysis been performed appropriately and rigorously? 

Reviewer #1: Yes

Reviewer #3: I Don't Know

4. Have the authors made all data underlying the findings in their manuscript fully available?

Reviewer #1: No

Reviewer #3: No

5. Is the manuscript presented in an intelligible fashion and written in standard English?

Reviewer #1: Yes

Reviewer #3: Yes

6. Review Comments to the Author

Reviewer #1: Dear Authors,

I still have some concerns related to your manuscript:

1. In my opinion only comparison of the whole control group to the all members of the experimental group is the prove of successful intervention, Does Fig 1 shows all members of alpha group or only responders ?

2. There is no point to Investigate whole brain EEG of responders if you do not compare it to non-responders, it does not explain the difference between the two

3. In the questions about other bands I was asking about possible training effects - so to say to compare them between control and experimental groups

4. All methodological explanations should be included in methods section

Reviewer #3: The authors used a 12-session alpha Neurofeedback training to induce high-alpha activity in a group of responders. These individuals underwent a whole-head EEG recording and were asked to emulate the strategy used during neurofeedback training in order to produce high alpha activity. During this time, high alpha bursts were used to apply EEG source localisation to find sources of NF-training induced alpha power changes. I believe that the study design underlies some major confounds and that this conclusion cannot be drawn so easily and that these limitations must be clearly addressed in the discussion.

1. Systematic differences in study design between groups, unrelated to the condition differences

a. Alpha group received different instructions than control group (mental imagery, etc.)

b. Alpha group received verbal feedback of successful alpha events (verbal feedback has a strong motivational effect, which can strongly influence learning processes). In case the control group received the same verbal feedback for successful events in their frequency band, this point can be ignored.

c. “not well trained” participants were removed from alpha group, but no such treatment was applied to the control group. How many participants of the control group would have to be excluded applying the same criteria?

 This makes it impossible to conclude, that found differences between groups originate from NF training.

2. There is no proof that the localization of alpha power is related to NF training.

a. There is no comparison between high alpha burst events in a baseline period before any training happened concerning their dipoles reconstructions and successful alpha events after NF training. It is entirely possible they would be the same and that NF training simply increases already present alpha oscillations, targeted by electrodes over the training site.

b. The authors did not present a comparison between the fixation period of the whole-head EEG and the periods, where participants had to copy their NF strategies. If there is no difference overall in alpha power between both periods, how can be inferred that this alpha activity has anything to do with NF training? What if the same source localisation strategy would be applied to high alpha bursts in the fixation period? If it was different, that would be more convincing. How many successful alpha events would there be in the fixation period, if the same criteria (1.5-fold higher amplitude than average of all the 1-s fixation EEGs) was applied?

3. The source localisation cannot be generalised for all NF alpha trainings

a. Alpha power reported in previous studies, generally increased over sites it was trained on. In this study, a general inference is made of where training-induced alpha changes occur, but it is evident that this is heavily dependent on training site.

4. Insufficient electrode numbers and missing T1 scans introduce the possibility of heavy mis-localization errors (up to 7 cm).

it is necessary that each of these limitations are clearly communicated in the discussion.

Some further details that need adjustment are outlined below

1. Previous studies have shown that training-induced alpha activity appeared diversely over the parieto-occipital [2] , fronto-parietal [3], or frontal regions [4] through topographic EEG analysis … Therefore, estimating the properties of the internal localized sources of the topography was a valuable way to explore generators of training-induced alpha activity.

The first two studies only show that alpha power increased over exactly the sites that were trained on (the first study used P3, P4, Pz, O1 and O2 as training sites, and the second study used F3, Fz, F4, P3, Pz and P4 as training sites). In the third study the authors cite, alpha power increase was trained over POz in 2 participants and was successful over POz. In three additional participants, the frequency of alpha oscillations was trained to be sped up. These 3 participants showed increased frequency of alpha oscillations over frontal areas, that means a faster alpha rhythm. This did not refer to power changes.

Therefore, it is difficult to argue that in this study, the authors explore the generators of training-induced alpha activity. These studies clearly show that training-induced alpha power changes depend on training site. The authors can therefore only conclude results for the specified training site.

2. Before NFT, researchers provided constructive strategies for successful alpha activity training reported by participants in our previous study, for example, pleasant or relaxing situations such as reading or wandering

this is a confound which needs to be discussed. This means that the instructions between conditions was not equal. It means that these strategies could have led to alpha increase independent of Neurofeedback training or EEG. It would only be no problem, if the control group received the same instructions.

3. In the resting period, the researchers use information on the cumulative waveform, e.g., the timestamps of high EEG amplitudes to help participants recall what kind of strategy they used to achieve a high amplitude.

was the same feedback given for the control group? Verbal feedback can have a tremendously motivating and positive effect on learning, which again is a confound for the NF training compared to control.

4. The 4-Hz bandwidth was randomly selected from the range of 7- 20 Hz. Thus, the Ctrl group received various kinds of 4-Hz amplitudes during each session [16].

how many participants in the control group received feedback including 10 Hz in their training range?

5. The experimental procedure of the whole-head EEG recording involved 5 runs. Each run interleaved four fixation and four alpha blocks (50 s each). In a fixation block, the “responder” was asked to keep his or her eyes open and not to engage in any training event. In an alpha block, the “responder” was asked to produce training-induced alpha activity. There was a 5-min rest period between two consecutive runs.

As noted above, I find it important to report, whether the fixation and the alpha increase blocks different from each other in alpha power overall (not just looking at successful alpha events). If not, can the authors be certain their findings refer to the dipoles of NF training induced alpha power changes? Or just high alpha power events (which also happen naturally). Is there here found dipole localization any different to what has been found for resting state alpha oscillations?

6. All of the successful alpha events contaminated with blinks, eye-movement artifacts (> 65 μV) or remarkable muscle activity were automatically removed and were further processed through four different averaging methods.

I assume that the authors mean the remaining events were further processed, because it sounds as if the removed ones were processed.

7. Six participants in the Alpha group who were not well trained were excluded from further analyses.

please state the exact exclusion criteria. Not well trained is very vague. If the authors would apply the same criterium to the ctrl group, how many participants would be excluded?

8. How many trials were average for each method? Please state the min, mean and max in the text.

9. The electrode number is still a problem and should be discussed. The authors refer to the paper from Rodin & Rodin, 1995, where a source localization was conducted with 19 electrodes. There is no proof in this paper, that this source localization was valid.

The second paper that is referred to by Sohrapour refers to the effect of electrode number on epileptic source localisation. Importantly, epileptic activity cannot be compared to healthy brain activity, as the strength of the activity largely exceeds normal activity, reducing error margin of source localization. Additionally, the patients studied in this particular study had seizures that strong, intracranial electrodes were implanted and surgery was conducted. Furthermore, in this study MRI scans of the individual participants were applied (which is not the case in the study under review). Also, the Sohrapour et. al find similar results for the source localisation, however the worst localisation is achieved using 32 electrodes. Excerpt from the paper: “In other words, the most dramatic decrease in localization error can be seen when going from 32 electrodes to 64 electrodes. The average localization error improves by 4 mm when going from 32 electrodes to 64 electrodes and improves by 1.3 mm when going from 64 electrodes to 96 electrodes,…

Additionally to the computer simulation I cited before, I also want to refer to a review paper, elaborating on EEG source imaging by Michel & Brunet (https://doi.org/10.3389/fneur.2019.00325):

“What is the minimal number of electrodes needed for reliable source localization? This question is often asked, particularly from the clinical community that intends to apply EEG source localization to the EEG that is routinely recorded with the standard 10-20 system, i.e., with only 19 electrodes. Several studies have demonstrated that this low number not only leads to blurring of the solution, but also to incorrect localization (49) compared the effective spatial resolution of different electrode montages (19-129 electrodes) and concluded that “the smallest topographic feature that can be resolved accurately by a 32-channel array is 7 cm in diameter, or about the size of a lobe of the brain.” All of this refers to source localisation in accordance with T1 scans.

Therefore, I believe it is not enough to state that the signal to noise ratio could be improved using T1 scans and higher density EEG, but to discuss as a limitation of the study, that the low-density EEG used introduces blurring and localization error, which in some cases can be substantial.

7. PLOS authors have the option to publish the peer review history of their article (what does this mean?). If published, this will include your full peer review and any attached files.

Reviewer #1: No

Reviewer #3: No

---

## [Author Response · Author response to Decision Letter 1]

7 Jul 2021

Dear Dr. Shaw,

Thank you for submitting your manuscript to PLOS ONE. After careful consideration, we feel that it has merit but does not fully meet PLOS ONE’s publication criteria as it currently stands. Therefore, we invite you to submit a revised version of the manuscript that addresses all the points raised during the review process.

In particular, I require that you provide full analyses including the non-responders.

Response: Thank you for the editor’s suggestion. In this study, NFT was conducted for all (sixty-three) participants. According to the criteria of “responder” (lines 167-169), six non-well-trained participants (“non-responders”) of the Alpha group were identified and excluded from further analyses, e. g., whole-head EEG data, EEG mapping and dipole source localization. In other words, only well-trained “responders” in the Alpha group participated in whole-head EEG recording, but not “non-responders”. Therefore, there are no analyses of whole-head EEG data, EEG mapping and dipole source localization for “non-responders”. In this revised version, we have added the comparison of the Ctrl group to the all participants of the Alpha groups in alpha amplitude (lines 267-275) and the amplitudes of bands (lines 284-293). The statistics have been added in the EEG data for NFT part of the “Results” section. The analysis of the alpha amplitude and successful alpha events between the fixation and alpha blocks were also added in the “Whole-head EEG data” part of the “Results” section (lines 303-311). 

Reviewer #1: 

Dear Authors,

I still have some concerns related to your manuscript:

1. In my opinion only comparison of the whole control group to the all members of the experimental group is the prove of successful intervention, Does Fig 1 show all members of alpha group or only responders?

Response: Thank you for the reviewer’s suggestion. The Fig 1 and Fig 2 show only responders of the Alpha group in the previous version. In this version, we have changed Fig 1 and Fig 2 which show all members of the Alpha (thirty-five participants) and Ctrl groups. Besides, the statistics have been added in the EEG data for NFT part of the “Results” section.

(lines 267-275) “The mean relative alpha amplitude progressively increased throughout the 12 sessions in the Alpha group but not in the Ctrl group (Fig. 1). ANOVA revealed significant main effects of group (F1,61 = 16.382, p < 0.001), session (F11,671 = 5.726, p < 0.001), and their interaction (F11,671 = 4.642, p < 0.001). The mean relative alpha amplitude showed no significant difference in the Ctrl group throughout NFT. In contrast, the mean relative alpha amplitudes of the 5th-12th sessions in the Alpha group significantly differed from those of the 1st session. In addition, a linear increase was observed in the Alpha group (R 2 = 0. 103, p < 0.001).”

(lines 284-293) “For the delta, theta and beta bands, ANOVA revealed no significant main effect of group (delta, F1,61 = 3.281, p = 0.075; theta, F1,61 = 2.465, p = 0.122; beta, F1,61 = 0.644, p = 0.425), session (delta, F1,61 = 1.427, p = 0.237; theta, F1,61 = 3.228, p = 0.077; beta, F1,61 = 1.166, p = 0.285), and their interaction (delta, F1,61 = 0.002, p = 0.966; theta, F1,61 = 0.230, p = 0.634; beta, F1,61 = 0.488, p = 0.488). In contrast, for the alpha band, significant main effects of group (F1,61 = 4.535, p = 0.037), session (F1,61 = 20.530, p < 0.001), and their interaction (F1,61 = 4.466, p = 0.039). The alpha band showed a significant difference between the two groups in the 12th session, but not in the 1st session. A significant difference between the 1st and 12th sessions in the alpha group was also observed.”

2. There is no point to Investigate whole brain EEG of responders if you do not compare it to non-responders, it does not explain the difference between the two

Response: Thank you for the reviewer’s suggestion. “Responders” could successfully induce alpha activity to provide stable and reliable alpha activity, at least more effectively than “non-responders”. In other words, “responders” could provide more successful alpha epoch than “non-responders” which is helpful for the dipole source analyze. Therefore, only “responders” participated in whole-head EEG recording, but not “non-responders”. In order to reveal that “responder” had learned to enhance training-induce alpha activity, in this revise version, we have added the analysis of the alpha amplitude and successful alpha events between the fixation and alpha blocks. The results showed that both parameters were significant different (lines 300-306) indicated that “responder” had learned how to control and enhance alpha activity after NFT. The purpose of this study is to investigate the sources of training-induced alpha activity. Therefore, we did not compare and explain the difference between the two. This is another interesting issue for future NFT research.

(lines 303-309) “The alpha amplitude of the alpha (13.02 ± 1.01) and fixation (10.01 ± 0.71) blocks were significant different (t = 4.822, p < 0.001). Besides, a significant difference between successful alpha events of the alpha (324.24 ± 3.97, range: 224-606) and fixation blocks (55.03 ± 22.27, range: 11-89) was also observed (t = 12.53, p < 0.001). These data indicated that “responders” had learned to successfully induce alpha activity in the alpha block. The successful alpha events were further processed through four different averaging methods.”

3. In the questions about other bands I was asking about possible training effects - so to say to compare them between control and experimental groups

Response: Thank you for the reviewer’s suggestion. For the delta, theta and beta bands, no significant difference was observed between the Ctrl and Alpha groups, and 1st and 12th sessions. In contrast, for the alpha band, significant differences were observed between the two groups in the 12th session, and between the 1st and 12th sessions in the Alpha group. The results were the same as the previous results indicated that only the amplitude of alpha band was increased after NFT but not the other bands. The detail statistics have been added in the EEG data for NFT part of the “Results” section.

(lines 284-293) “For the delta, theta and beta bands, ANOVA revealed no significant main effect of group (delta, F1,61 = 3.281, p = 0.075; theta, F1,61 = 2.465, p = 0.122; beta, F1,61 = 0.644, p = 0.425), session (delta, F1,61 = 1.427, p = 0.237; theta, F1,61 = 3.228, p = 0.077; beta, F1,61 = 1.166, p = 0.285), and their interaction (delta, F1,61 = 0.002, p = 0.966; theta, F1,61 = 0.230, p = 0.634; beta, F1,61 = 0.488, p = 0.488). In contrast, for the alpha band, significant main effects of group (F1,61 = 4.535, p = 0.037), session (F1,61 = 20.530, p < 0.001), and their interaction (F1,61 = 4.466, p = 0.039). The alpha band showed a significant difference between the two groups in the 12th session, but not in the 1st session. A significant difference between the 1st and 12th sessions in the alpha group was also observed.”

4. 4. All methodological explanations should be included in methods section

Response: Thank you for reviewer’s suggestion. We have added the methodological explanations in the “Methods” section, e.g., the description of the trainer (lines 113-115), the benefit of the bipolar recording (lines 131-132), the 3D digitizer for individual electrode positions (lines 174-175), the calculation of ERSPA (lines 205-208) and the ICA for dipole source localization (lines 221-228).

Reviewer #3: 

The authors used a 12-session alpha Neurofeedback training to induce high-alpha activity in a group of responders. These individuals underwent a whole-head EEG recording and were asked to emulate the strategy used during neurofeedback training in order to produce high alpha activity. During this time, high alpha bursts were used to apply EEG source localisation to find sources of NF-training induced alpha power changes. I believe that the study design underlies some major confounds and that this conclusion cannot be drawn so easily and that these limitations must be clearly addressed in the discussion.

1. Systematic differences in study design between groups, unrelated to the condition differences

a. Alpha group received different instructions than control group (mental imagery, etc.)

Response: Thank you for the reviewer’s suggestion. Both Ctrl and Alpha groups received the same instructions.

b. Alpha group received verbal feedback of successful alpha events (verbal feedback has a strong motivational effect, which can strongly influence learning processes). In case the control group received the same verbal feedback for successful events in their frequency band, this point can be ignored.

Response: Thank you for the reviewer’s suggestion. Both Ctrl and Alpha groups received same verbal feedback. 

c. “not well trained” participants were removed from alpha group, but no such treatment was applied to the control group. How many participants of the control group would have to be excluded applying the same criteria?  This makes it impossible to conclude, that found differences between groups originate from NF training.

Response: Thank you for the reviewer’s suggestion. In this study, NFT was conducted for all (sixty-three) participants we recruited. According to the criteria of “responder” (lines 167-169), six non-well-trained participants of the Alpha group were identified and excluded from further analyses, e. g., whole-head EEG data, EEG mapping and dipole source localization. We have modified the description to the last paragraph of the “EEG data for NFT” part of the “Results” section.

(lines 299-301) “Twenty-nine participants (83%) were identified as “responders” in this study. Therefore, six participants in the Alpha group who were not well trained were excluded from whole-head EEG data, EEG mapping and dipole source localization analyses.” 

2. There is no proof that the localization of alpha power is related to NF training.

a. There is no comparison between high alpha burst events in a baseline period before any training happened concerning their dipoles reconstructions and successful alpha events after NF training. It is entirely possible they would be the same and that NF training simply increases already present alpha oscillations, targeted by electrodes over the training site.

Response: Thank you for the reviewer’s constructive suggestion. The purpose of NFT is to enable an individual to learn to self-regulate specific frequency band(s) which already exist in brain activity. Therefore, it is possible that NFT simply increases already present alpha oscillations. In this study, the Alpha group showed successful training of the alpha activity after NFT, but not the Ctrl group. In addition, in order to reveal that participants had learned to enhance training-induce alpha activity, the whole-head EEG was recorded after NFT. The results showed that the alpha amplitude and successful alpha events between the fixation and alpha blocks were significant different indicated that participants had learned how to control and enhance alpha activity after NFT. These results have been added in the “Whole-head EEG data” part of the “Results” section.

(lines 303-309) “The alpha amplitude of the alpha (13.02 ± 1.01) and fixation (10.01 ± 0.71) blocks were significant different (t = 4.822, p < 0.001). Besides, a significant difference between successful alpha events of the alpha (324.24 ± 3.97, range: 224-606) and fixation blocks (55.03 ± 22.27, range: 11-89) was also observed (t = 12.53, p < 0.001). These data indicated that “responders” had learned to successfully induce alpha activity in the alpha block. The successful alpha events were further processed through four different averaging methods.”

b. The authors did not present a comparison between the fixation period of the whole-head EEG and the periods, where participants had to copy their NF strategies. If there is no difference overall in alpha power between both periods, how can be inferred that this alpha activity has anything to do with NF training? What if the same source localisation strategy would be applied to high alpha bursts in the fixation period? If it was different, that would be more convincing. How many successful alpha events would there be in the fixation period, if the same criteria (1.5-fold higher amplitude than average of all the 1-s fixation EEGs) was applied?

Response: Thank you for the reviewer’s constructive suggestion. In this revised version, we have compared the alpha amplitude and successful alpha events between the fixation and alpha blocks. Both parameters showed significant differences between the alpha and fixation blocks. These results have been added in the “Whole-head EEG data” part of the “Results” section.

(lines 303-309) “The alpha amplitude of the alpha (13.02 ± 1.01) and fixation (10.01 ± 0.71) blocks were significant different (t = 4.822, p < 0.001). Besides, a significant difference between successful alpha events of the alpha (324.24 ± 3.97, range: 224-606) and fixation blocks (55.03 ± 22.27, range: 11-89) was also observed (t = 12.53, p < 0.001). These data indicated that “responders” had learned to successfully induce alpha activity in the alpha block. The successful alpha events were further processed through four different averaging methods.” 

3. The source localisation cannot be generalised for all NF alpha trainings

a. Alpha power reported in previous studies, generally increased over sites it was trained on. In this study, a general inference is made of where training-induced alpha changes occur, but it is evident that this is heavily dependent on training site.

Response: Thank you for the reviewer’s constructive suggestion. We have modified the description in the “Discussion” section.

(lines 486-488) “our findings provide valuable information for future studies that three dipoles could be used for dipole source analysis of training-induced alpha activity, especially the training sites are around the central regions.”

4. Insufficient electrode numbers and missing T1 scans introduce the possibility of heavy mis-localization errors (up to 7 cm). It is necessary that each of these limitations are clearly communicated in the discussion.

Response: Thank you for the reviewer’s suggestion. The low-density EEG used in dipole source localization may cause blurring and localization error. However, low-density electrodes used in dipole source localization may obtain valuable insight in application with some brain activities, such as epileptic spikes. Although the amplitude of alpha activity may not greater than that of epileptic activity, we used average method to increase SNR of alpha activity. A good SNR can decrease the estimation error of dipole source reconstruction. Besides, Fuchs et al. (2002) (https://doi.org/10.1016/S1388-2457(02)00030-5) reported that no significant differences in source localization were observed and showed the similarity in source localization between individual and standardized BEM models. The evidence revealed that standardized BEM models we used in this study may remain sufficient for source localization. A provided source solution remains an estimation that depends on priori assumptions which can be neurophysiological, biophysical and anatomical knowledge about the distribution of neuronal activity. However, there are still no assumptions about training-induced alpha activity to valid our findings. In the present study, we found dipoles approximately clustered in the bilateral precuneus, PCC and middle temporal regions. Of course, the validation of these findings should be tested in the future work, e.g., fMRI. We have modified the discussion of limitations in the “Discussion” section.

(lines 529-542) “One of the limitations of the present study was low-density EEG. The low-density EEG used in dipole source localization may cause blurring and localization error. However, low-density EEG (<32 electrodes) used in dipole source localization may obtain valuable insight in application with some brain activities, such as epileptic spikes [44]. Although the amplitude of alpha activity may not greater than that of epileptic activity, we used average method to increase SNR of alpha activity. A good SNR can decrease the estimation error of dipole source localization. Indeed, more electrodes could reduce more distance errors in dipole source analysis [45]. More electrodes, such as 64 electrodes or more, should be tested in future research. The other limitation was absence of individual anatomical images. Individual anatomical image can improve the source localization accuracy. However, no significant differences in source localization were observed and showed the similarity in source localization between individual and standardized BEM models [46]. The evidence revealed that standardized BEM models used in this study may remain sufficient.”

Some further details that need adjustment are outlined below

1. Previous studies have shown that training-induced alpha activity appeared diversely over the parieto-occipital [2], fronto-parietal [3], or frontal regions [4] through topographic EEG analysis … Therefore, estimating the properties of the internal localized sources of the topography was a valuable way to explore generators of training-induced alpha activity.

The first two studies only show that alpha power increased over exactly the sites that were trained on (the first study used P3, P4, Pz, O1 and O2 as training sites, and the second study used F3, Fz, F4, P3, Pz and P4 as training sites). In the third study the authors cite, alpha power increase was trained over POz in 2 participants and was successful over POz. In three additional participants, the frequency of alpha oscillations was trained to be sped up. These 3 participants showed increased frequency of alpha oscillations over frontal areas, that means a faster alpha rhythm. This did not refer to power changes.

Therefore, it is difficult to argue that in this study, the authors explore the generators of training-induced alpha activity. These studies clearly show that training-induced alpha power changes depend on training site. The authors can therefore only conclude results for the specified training site.

Response: Thank you for the reviewer’s constructive suggestion. We have modified the description in the “Discussion” section.

(lines 486-488) “our findings provide valuable information for future studies that three dipoles could be used for dipole source analysis of training-induced alpha activity, especially the training sites are around the central regions.”

2. Before NFT, researchers provided constructive strategies for successful alpha activity training reported by participants in our previous study, for example, pleasant or relaxing situations such as reading or wandering

this is a confound which needs to be discussed. This means that the instructions between conditions was not equal. It means that these strategies could have led to alpha increase independent of Neurofeedback training or EEG. It would only be no problem, if the control group received the same instructions.

Response: Thank you for the reviewer’s suggestion. Both Ctrl and Alpha groups received the same instructions.

3. In the resting period, the researchers use information on the cumulative waveform, e.g., the timestamps of high EEG amplitudes to help participants recall what kind of strategy they used to achieve a high amplitude.

was the same feedback given for the control group? Verbal feedback can have a tremendously motivating and positive effect on learning, which again is a confound for the NF training compared to control.

Response: Thank you for the reviewer’s suggestion. Both Ctrl and Alpha groups received the same verbal feedback in the resting period.

4. The 4-Hz bandwidth was randomly selected from the range of 7-20 Hz. Thus, the Ctrl group received various kinds of 4-Hz amplitudes during each session [16].

how many participants in the control group received feedback including 10 Hz in their training range?

Response: Thank you for the reviewer’s suggestion. The Ctrl group received amplitude of 4-Hz bandwidth as feedback. The 4-Hz bandwidth was randomly selected second-by-second from the range of 7-20 Hz during training block. In other words, the 4-Hz bandwidth was different between each second during training block for each participants of the Ctrl group. Therefore, all participants of the Ctrl group were possibly to receive feedback including 10 Hz. We have modified the description in the “Neurofeedback training and processing” part of the “Materials and methods” section.

(lines 144-146) “The 4-Hz bandwidth was randomly selected second-by-second from the range of 7-20 Hz. Thus, the Ctrl group received various kinds of 4-Hz amplitudes during each training block.”

5. The experimental procedure of the whole-head EEG recording involved 5 runs. Each run interleaved four fixation and four alpha blocks (50 s each). In a fixation block, the “responder” was asked to keep his or her eyes open and not to engage in any training event. In an alpha block, the “responder” was asked to produce training-induced alpha activity. There was a 5-min rest period between two consecutive runs.

As noted above, I find it important to report, whether the fixation and the alpha increase blocks different from each other in alpha power overall (not just looking at successful alpha events). If not, can the authors be certain their findings refer to the dipoles of NF training induced alpha power changes? Or just high alpha power events (which also happen naturally). Is there here found dipole localization any different to what has been found for resting state alpha oscillations?

Response: Thank you for the reviewer’s suggestion. In this revised version, we have compared the alpha amplitude and successful alpha events between the fixation and alpha blocks. Both parameters showed significant differences between the alpha and fixation blocks. These results have been added in the “Whole-head EEG data” part of the “Results” section.

(lines 303-311) “The alpha amplitude of the alpha (13.02 ± 1.01) and fixation blocks (10.01 ± 0.71) were significant different (t = 4.822, p < 0.001). Besides, a significant difference between successful alpha events of the alpha (324.24 ± 3.97, range: 224-606) and fixation blocks (55.03 ± 22.27, range: 11-89) was also observed (t = 12.53, p < 0.001). These data indicated that “responders” had learned to successfully induce alpha activity in the alpha block. The successful alpha events were further processed through four different averaging methods.”

6. All of the successful alpha events contaminated with blinks, eye-movement artifacts (> 65 μV) or remarkable muscle activity were automatically removed and were further processed through four different averaging methods.

I assume that the authors mean the remaining events were further processed, because it sounds as if the removed ones were processed.

Response: Thank you for the reviewer’s suggestion. We have corrected the description in the “Whole-head EEG and processing” part of the “Materials and methods” section.

(lines 188-192) “The successful alpha events contaminated with blinks, eye-movement artifacts (> 65 μV) or remarkable muscle activity were automatically removed. After that, all of the remaining successful alpha events were further processed through four different averaging methods.”

7. Six participants in the Alpha group who were not well trained were excluded from further analyses.

please state the exact exclusion criteria. Not well trained is very vague. If the authors would apply the same criterium to the ctrl group, how many participants would be excluded?

Response: Thank you for the reviewer’s suggestion. In this study, NFT was conducted for all (sixty-three) participants. According to the criteria of “responder” (lines 167-169), six non-well-trained participants of the Alpha group were identified and excluded from further analyses, e. g., whole-head EEG data, EEG mapping and dipole source localization. No participant of the Ctrl group was identified as “responder”. We have modified the description to the last paragraph of the EEG data for NFT part of the “Results” section.

(lines 299-301) “Twenty-nine participants (83%) were identified as “responders” in this study. Therefore, six participants in the Alpha group who were not well trained were excluded from whole-head EEG data, EEG mapping and dipole source localization analyses.”

8. How many trials were average for each method? Please state the min, mean and max in the text.

Response: Thank you for the reviewer’s suggestion. We have added the statistic in the “Whole-head EEG data” part of the “Results” section.

(lines 309-311) “Finally, alpha epochs of the MPA and ERSPA (324.24 ± 22.27, range: 224-606), PA (186.03 ± 14.27, range: 66-358) and NA (138.21 ± 10.18, rang: 61-285) methods were identified and averaged.”

9. The electrode number is still a problem and should be discussed. The authors refer to the paper from Rodin & Rodin, 1995, where a source localization was conducted with 19 electrodes. There is no proof in this paper, that this source localization was valid. 

The second paper that is referred to by Sohrapour refers to the effect of electrode number on epileptic source localisation. Importantly, epileptic activity cannot be compared to healthy brain activity, as the strength of the activity largely exceeds normal activity, reducing error margin of source localization. Additionally, the patients studied in this particular study had seizures that strong, intracranial electrodes were implanted and surgery was conducted. Furthermore, in this study MRI scans of the individual participants were applied (which is not the case in the study under review). Also, the Sohrapour et. al find similar results for the source localisation, however the worst localisation is achieved using 32 electrodes. Excerpt from the paper: “In other words, the most dramatic decrease in localization error can be seen when going from 32 electrodes to 64 electrodes. The average localization error improves by 4 mm when going from 32 electrodes to 64 electrodes and improves by 1.3 mm when going from 64 electrodes to 96 electrodes,…Additionally to the computer simulation I cited before, I also want to refer to a review paper, elaborating on EEG source imaging by Michel & Brunet (https://doi.org/10.3389/fneur.2019.00325):“What is the minimal number of electrodes needed for reliable source localization? This question is often asked, particularly from the clinical community that intends to apply EEG source localization to the EEG that is routinely recorded with the standard 10-20 system, i.e., with only 19 electrodes. Several studies have demonstrated that this low number not only leads to blurring of the solution, but also to incorrect localization (49) compared the effective spatial resolution of different electrode montages (19-129 electrodes) and concluded that “the smallest topographic feature that can be resolved accurately by a 32-channel array is 7 cm in diameter, or about the size of a lobe of the brain.” All of this refers to source localisation in accordance with T1 scans.

Therefore, I believe it is not enough to state that the signal to noise ratio could be improved using T1 scans and higher density EEG, but to discuss as a limitation of the study, that the low-density EEG used introduces blurring and localization error, which in some cases can be substantial.

Response: Thank you for the reviewer’s suggestion. The low-density EEG used in dipole source localization may cause blurring and localization error. However, low-density electrodes used in dipole source localization may obtain valuable insight in application with some brain activities, such as epileptic spikes. Although the amplitude of alpha activity may not greater than that of epileptic activity, we used average method to increase SNR of alpha activity. A good SNR can decrease the estimation error of dipole source reconstruction. Besides, Fuchs et al. (2002) (https://doi.org/10.1016/S1388-2457(02)00030-5) reported that no significant differences in source localization were observed and showed the similarity in source localization between individual and standardized BEM models. The evidence revealed that standardized BEM models we used in this study may remain sufficient for source localization. The provided source solution remains an estimation that depends on priori assumptions which can be neurophysiological, biophysical and anatomical knowledge about the distribution of neuronal activity. However, there are still no assumptions about training-induced alpha activity to valid our findings. In the present study, we found dipoles approximately clustered in the bilateral precuneus, PCC and middle temporal regions. Of course, the validation of these findings should be tested in the future work, e.g., fMRI. We have modified the discussion of limitations in the “Discussion” section.

(lines 529-542) “One of the limitations of the present study was low-density EEG. The low-density EEG used in dipole source localization may cause blurring and localization error. However, low-density EEG (<32 electrodes) used in dipole source localization may obtain valuable insight in application with some brain activities, such as epileptic spikes [44]. Although the amplitude of alpha activity may not greater than that of epileptic activity, we used average method to increase SNR of alpha activity. A good SNR can decrease the estimation error of dipole source localization. Indeed, more electrodes could reduce more distance errors in dipole source analysis [45]. More electrodes, such as 64 electrodes or more, should be tested in future research. The other limitation was absence of individual anatomical images. Individual anatomical image can improve the source localization accuracy. However, no significant differences in source localization were observed and showed the similarity in source localization between individual and standardized BEM models [46]. The evidence revealed that standardized BEM models used in this study may remain sufficient.”

---

## [Decision Letter · Decision Letter 2]

29 Jul 2021

PONE-D-20-20804R2

Equivalent current dipole sources of neurofeedback training-induced alpha activity through temporal/spectral analytic techniques.

PLOS ONE

Dear Dr. Shaw,

Thank you for submitting your manuscript to PLOS ONE. After careful consideration, we feel that it has merit but does not fully meet PLOS ONE’s publication criteria as it currently stands. Therefore, we invite you to submit a revised version of the manuscript that addresses the points raised by the Reviewer 1 during the review process.

We look forward to receiving your revised manuscript.

Kind regards,

Francesco Di Russo, Ph.D.

Academic Editor

PLOS ONE

Journal Requirements:

Reviewers' comments:

Reviewer's Responses to Questions

**Comments to the Author**

1. If the authors have adequately addressed your comments raised in a previous round of review and you feel that this manuscript is now acceptable for publication, you may indicate that here to bypass the “Comments to the Author” section, enter your conflict of interest statement in the “Confidential to Editor” section, and submit your "Accept" recommendation.

Reviewer #1: All comments have been addressed

Reviewer #3: (No Response)

2. Is the manuscript technically sound, and do the data support the conclusions?

Reviewer #1: Partly

Reviewer #3: No

3. Has the statistical analysis been performed appropriately and rigorously? 

Reviewer #1: Yes

Reviewer #3: Yes

4. Have the authors made all data underlying the findings in their manuscript fully available?

Reviewer #1: Yes

Reviewer #3: Yes

5. Is the manuscript presented in an intelligible fashion and written in standard English?

Reviewer #1: Yes

Reviewer #3: Yes

6. Review Comments to the Author

Reviewer #1: Please specify type of post hoc tests and their statistics used in the results sections "EEG data for NFT" - does it confirm F test results ? if so for which data points ? Please provide statistics for full groups in Fig 1. and Fig2 and accordingly corrects figures

Also please also cite the literature behind your statements "The benefit of the bipolar recording was to reduce the possible artifacts of motion or eye blinks" - I did not find any example. Instead bipolar setting is used to monitor eye blinks - please check: https://doi.org/10.1007/s10548-019-00707-x

Reviewer #3: (No Response)

7. PLOS authors have the option to publish the peer review history of their article (what does this mean?). If published, this will include your full peer review and any attached files.

Reviewer #1: No

Reviewer #3: No

---

## [Author Response · Author response to Decision Letter 2]

12 Aug 2021

Reviewer #1: 

1. Please specify type of post hoc tests and their statistics used in the results sections "EEG data for NFT" - does it confirm F test results? if so for which data points? Please provide statistics for full groups in Fig 1. and Fig2 and accordingly corrects figures.

Response: Thank you for the reviewer’s suggestion. In this study, two-factor mixed analysis of variance (ANOVA) followed by Bonferroni post hoc testing was conducted on the mean relative alpha amplitude and amplitudes of the delta (1-3 Hz), theta (4-7 Hz), alpha (8-12 Hz), and beta (13-30 Hz) bands. The statistics in the "EEG data for NFT" are the results of the two-factor ANOVA followed by Bonferroni post hoc testing. We have modified the description of the type of post hoc test and statistic in the “Statistical analyses” part of the “Materials and methods” section.

(lines 244-247) “To evaluate the alpha NFT course and spectra, two-factor mixed analysis of variance (ANOVA) followed by Bonferroni post hoc testing was conducted on the mean relative alpha amplitude and amplitudes of the delta (1-3 Hz), theta (4-7 Hz), alpha (8-12 Hz), and beta (13-30 Hz) bands.”

2. Also please also cite the literature behind your statements "The benefit of the bipolar recording was to reduce the possible artifacts of motion or eye blinks" - I did not find any example. Instead bipolar setting is used to monitor eye blinks - please check: https://doi.org/10.1007/s10548-019-00707-x

Response: Thank you for the reviewer’s suggestion. The bipolar recordings are used to monitor focal brain activity (Schomer D. L. & Da Silva, F. L. 2012). Bipolar recordings by placing electrodes around the eyes is used to measure the electrooculogram (EOG), that is the electrical potential caused by blinks and eye movements. Therefore, in this study, we used bipolar recording to monitor the focal brain activity around the central region. The bipolar recording can reduce any external artifact, such as movement artifacts or eye blinks (Marzbani et al., 2016, page 145, the bottom left side). We have added the citation behind the statement (line 132).

References

1. Schomer DL, Da Silva FL. Niedermeyer's electroencephalography: basic principles, clinical applications, and related fields. Lippincott Williams & Wilkins. 2012.

2. Marzbani H, Marateb HR, Mansourian M. Neurofeedback: a comprehensive review on system design, methodology and clinical applications. Basic Clin. Neurosci. 2016;7(2):143-58.

---

## [Decision Letter · Decision Letter 3]

11 Feb 2022

Equivalent current dipole sources of neurofeedback training-induced alpha activity through temporal/spectral analytic techniques.

PONE-D-20-20804R3

Dear Dr. Shaw,

We’re pleased to inform you that your manuscript has been judged scientifically suitable for publication and will be formally accepted for publication once it meets all outstanding technical requirements.

Kind regards,

Daqing Guo

Academic Editor

PLOS ONE

Additional Editor Comments (optional):

Reviewers' comments:

Reviewer's Responses to Questions

**Comments to the Author**

1. If the authors have adequately addressed your comments raised in a previous round of review and you feel that this manuscript is now acceptable for publication, you may indicate that here to bypass the “Comments to the Author” section, enter your conflict of interest statement in the “Confidential to Editor” section, and submit your "Accept" recommendation.

Reviewer #1: All comments have been addressed

Reviewer #5: All comments have been addressed

2. Is the manuscript technically sound, and do the data support the conclusions?

Reviewer #1: Partly

Reviewer #5: Yes

3. Has the statistical analysis been performed appropriately and rigorously? 

Reviewer #1: Yes

Reviewer #5: Yes

4. Have the authors made all data underlying the findings in their manuscript fully available?

Reviewer #1: No

Reviewer #5: Yes

5. Is the manuscript presented in an intelligible fashion and written in standard English?

Reviewer #1: Yes

Reviewer #5: Yes

6. Review Comments to the Author

Reviewer #1: I have no more questions assuming that the experimental group in figures 1 and 2 includes all participants including non-responders

Reviewer #5: (No Response)

7. PLOS authors have the option to publish the peer review history of their article (what does this mean?). If published, this will include your full peer review and any attached files.

Reviewer #1: No

Reviewer #5: No

---

## [Editor Report · Acceptance letter]

17 Feb 2022

PONE-D-20-20804R3 

Equivalent current dipole sources of neurofeedback training-induced alpha activity through temporal/spectral analytic techniques. 

Dear Dr. Shaw:

I'm pleased to inform you that your manuscript has been deemed suitable for publication in PLOS ONE. Congratulations! Your manuscript is now with our production department. 

Kind regards, 

on behalf of

Dr. Daqing Guo 

Academic Editor

PLOS ONE